# Elastic Robust Unlearning of Specific Knowledge in Large Language Models

**Yize Sui**[1], **Jing Ren**[1], **Wenjing Yang**[1], **Ruochun Jin**[1], **Liyang Xu**[1], **Xiyao Liu**[2], **Ji Wang**[1]*

[1]College of Computer Science and Technology,
National University of Defense Technology
[2]School of Computer Science and Engineering,
Central South University
{suiyize18, renjing, wenjing.yang, jinrc, xuliyang08, wj}@nudt.edu.cn
lxyzoewx@csu.edu.cn

## Abstract

LLM unlearning aims to remove sensitive or harmful information within the model, thus reducing the potential risk of generating unexpected information. However, existing Preference Optimization (PO)-based unlearning methods suffer two limitations. First, their rigid reward setting limits the effect of unlearning. Second, the lack of robustness causes unlearned information to reappear. To remedy these two weaknesses, we present a novel LLM unlearning optimization framework, namely Elastic Robust Unlearning (ERU), to efficiently and robustly remove specific knowledge from LLMs. We design the elastic reward setting instead of the rigid reward setting to enhance the unlearning performance. Meanwhile, we incorporate the refusal feature ablation into the unlearning process to trigger specific failure patterns for efficiently enhancing the robustness of the PO-based unlearning methods in multiple scenarios. Experimental results show that ERU can improve the unlearning effectiveness significantly while maintaining high utility performance. Especially, on the WMDP-Bio benchmark, ERU shows a 9% improvement over the second-best method, and maintains 83% performance even under 1,000 sample fine-tuned retraining attacks, significantly better than the baseline method.

## 1 Introduction

With the rapid development of Large Language Models (LLMs) [1–3], their potential applications across various fields have increasingly become evident. This potential primarily stems from their extensive pre-trained knowledge base and exceptional generalization capabilities. However, due to the possibility that training data may contain copyrighted content, personal privacy information, and harmful speech among other undesirable elements [4–6], LLMs inevitably absorb some negative behavioral patterns during learning. These negative behaviors not only pose a threat to information security but may also have adverse social impacts, thereby hindering the widespread application of LLMs in real-world scenarios. Therefore, ensuring that LLMs are aligned with human values and intentions is crucial for maintaining their credibility and safety.

Since retraining models to remove specific undesirable content is costly and time-consuming, researchers have explored methods including safe fine-tuning [7] and adversarial training (AT) [8, 9]. However, recent studies on interpretability [10], representation engineering [11, 12], and continual learning [13–15] indicate that surface safety fine-tuning is difficult to ensure that LLMs are harmless in all scenarios, because this approach is not sufficient to fundamentally change the knowledge and capabilities of LLMs. Given that undesirable responses often stem from the same harmful knowledge

---

*Corresponding author.

39th Conference on Neural Information Processing Systems (NeurIPS 2025).

[16], the concept of LLM unlearning has been proposed to directly remove such harmful knowledge from LLMs [17, 18]. Currently, the basic method of LLM unlearning is to use the Gradient Ascent (GA) strategy on the forget set to realize knowledge unlearning by reversing the optimization process of gradient descent. Recently, inspired by Direct Preference Optimization (DPO) [19], Negative Preference Optimization (NPO) [20] treats the forget set as negative preference data, thereby assigning a lower likelihood to unlearned knowledge. Unlike NPO, which only focuses on negative feedback, Alternative Preference Optimization (AltPO) [21] innovatively combines positive and negative feedback information to effectively solve the problem of decreasing model output quality.

Despite LLM unlearning has made significant progress in eliminating harmful behaviors in LLMs, it still exhibits notable limitations. Firstly, current PO-based excessively unlearning methods rely on rigid reward setting, which restricts the unlearning effectiveness to some extent [22]. Secondly, the unlearned models generated by existing methods often lack robustness, making them vulnerable to the threat of knowledge recovery. For example, unlearned models may be reactivated on unlearned dangerous knowledge by carefully crafted adversarial prompts [23] or contextual interactions [24]. In addition, Deeb and Roger [25] show that fine-tuning the unlearned model with only a small number of samples can effectively restore the removed knowledge.

In this work, we propose a robust LLM unlearning optimization framework named ERU (Elastic Robust Unlearning), which aims to effectively and robustly remove specific knowledge and build trustworthy LLMs. Specifically, ERU adopts the elastic reward setting instead of the traditional rigid reward setting, achieving a more flexible balance between the reference-based reward and the reference-free reward. Subsequently, the ERU formulates the robust unlearning process as a max-minimum optimization problem, where the inner loop is carried out by simulating worst-case perturbations through refusal feature ablation [26], and the outer loop focuses on removing harmful knowledge. We conduct extensive experiments on multiple LLM unlearning benchmarks such as RWKU [27], MUSE [28], TOFU [29], and WMDP [30], covering LLMs such as LLaMA-2-7B-Chat [3] and LLaMA-3-8B-Instruct [31], and the experimental results fully verify the superiority of the ERU framework.

Our contribution can be summarized as follows: **(1)** We propose a novel robust LLM unlearning optimization framework called ERU, which can efficiently and robustly remove specific knowledge from LLMs. **(2)** The elastic reward setting is designed to achieve a more flexible balance between reference-based reward and reference-free reward, enhancing the unlearning effectiveness while maintaining the model utility. **(3)** We incorporate the refusal feature ablation into the unlearning process, which efficiently improves the unlearning robustness of the ERU. **(4)** Extensive experiments on multiple unlearning benchmarks and models show that the proposed ERU method has significant advantages in terms of unlearning effectiveness, utility preservation, and unlearning robustness.

## 2 Preliminary

In this section, we present the necessary background and closely related prior work. Additional related work is discussed in Section A.

### 2.1 Threat Model

We assume the existence of a unlearned white-box unlearned LLM $\pi_\theta^\dagger$ with weights $\theta$, allowing adversaries to modify its weights or intervene in the activation space during inference. The adversary has full white-box access to $\pi_\theta^\dagger$ and attempts to recover the removed dangerous knowledge through specific methods under reasonable computational cost constraints. Our goal is to design a robust unlearning method $\mathcal{S}$ such that the performance of $\pi_\theta^\dagger$ processed by $\mathcal{S}$ on the forget objects is significantly reduced even if the adversary attempts to recover the forget objects, while ensuring that the utility performance of the model is not affected.

### 2.2 Problem Formulation

For a large language model $\pi_\theta$ trained on dataset $\mathcal{D} = \{(x_i, y_i) \mid i = 1, 2, ..., N\}$, we define the target distribution of knowledge to be unlearned as the forget set $\mathcal{D}_f \subseteq \mathcal{D}$. Machine unlearning aims to fine-tune $\pi_\theta$ to behave as if it had only been trained on the retain set $\mathcal{D}_R = \mathcal{D}/\mathcal{D}_f$. Considering the high cost of retraining the model on $\mathcal{D}_R$ from scratch, LLM unlearning has been widely studied

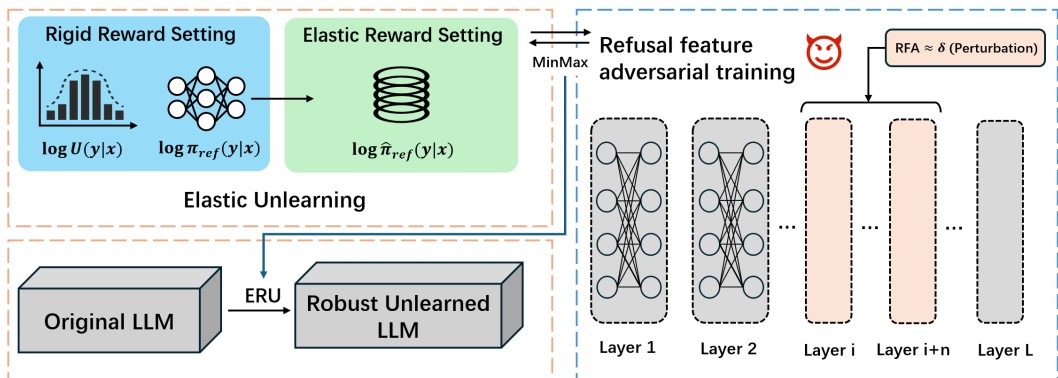

Figure 1: Overview of the ERU framework, which can implement robust LLM unlearning.

as an effective approximate unlearning method, which only relies on $\mathcal{D}_f$ and a part of the retain set ($\mathcal{D}_r \subseteq \mathcal{D}_R$). The problem of LLM unlearning can be transformed into a regularized optimization problem that balances the forget and retain objectives:

$$\min_{\boldsymbol{\theta}} \underbrace{\mathbb{E}_{(x_f, y_f) \in \mathcal{D}_f} \left[ \ell_f(y_f \mid x_f; \boldsymbol{\theta}) \right]}_{\text{Forget loss}} + \lambda \underbrace{\mathbb{E}_{(x_r, y_r) \in \mathcal{D}_r} \left[ \ell_r(y_r \mid x_r; \boldsymbol{\theta}) \right]}_{\text{Retain loss}} \tag{1}$$

where $\ell_f$ and $\ell_r$ represent forget and retain losses in generating the response $y$ given the input $x$, and $\lambda$ is a regularization parameter to balance them. Let $Acc_{\mathcal{D}_f}(\pi_\theta)$ and $Acc_{\mathcal{D}_r}(\pi_\theta)$ represent the inference accuracy of $\pi_\theta$ on $\mathcal{D}_f$ and $\mathcal{D}_r$, respectively. We expect unlearned LLM $\pi_\theta^\dagger$ to exhibit low $Acc_{\mathcal{D}_f}(\pi_\theta^\dagger)$ on $\mathcal{D}_f$ while maintaining high $Acc_{\mathcal{D}_r}(\pi_\theta^\dagger)$ on $\mathcal{D}_r$. However, the goal of potential adversaries is to reverse low $Acc_{\mathcal{D}_f}(\pi_\theta^\dagger)$. Therefore, a robust unlearned LLM is able to resist threats from adversaries.

### 2.3 From PO to LLM Unlearning

**Direct Preference Optimization (DPO).** DPO serves as a key prerequisite to incorporate LLM unlearning into the preference optimization framework. This method reparameterizes the following reward function $r(x, y)$ through a closed-form expression with the optimal policy instead of learning an explicit reward model:

$$r(x, y) = \beta \log \frac{\pi_\theta(y \mid x)}{\pi_{\text{ref}}(y \mid x)} + \beta \log Z(x), \tag{2}$$

where $Z(x)$ is the partition function, $\pi_\theta(y \mid x)$ represent the prediction probability of the LLM $\pi_\theta$ given the input-response pair $(x, y)$ and $\pi_{\text{ref}}$ is the reference model. By integrating the reward formula into the Bradley-Terry ranking objective [32], the DPO can express the probability of preference data with the policy model rather than the reward model. Therefore, the objective of the DPO is:

$$\mathcal{L}_{DPO}(\pi_\theta, \pi_{\text{ref}}) = -\mathbb{E}_{(x, y_w, y_l) \sim \mathcal{D}} \left[ \log \sigma \left( \beta \log \frac{\pi_\theta(y_w \mid x)}{\pi_{\text{ref}}(y_w \mid x)} - \beta \log \frac{\pi_\theta(y_l \mid x)}{\pi_{\text{ref}}(y_l \mid x)} \right) \right], \tag{3}$$

where $\sigma(\cdot)$ denotes the sigmoid function, $\beta > 0$ is the inverse temperature, and $(x, y_w, y_l)$ is the preference pair consisting of prompts, winning responses, and losing responses in the preference dataset $\mathcal{D}$.

**Negative Preference Optimization (NPO).** In the beginning, the LLM unlearning is implemented by gradient ascent (GA):

$$\mathcal{L}_{\text{GA}}(\pi_\theta) = -\underbrace{\mathbb{E}_{(x, y) \sim \mathcal{D}_f} \left[ -\log(\pi_\theta(y \mid x)) \right]}_{\text{prediction loss}}, \tag{4}$$

but this method always faces the risk of catastrophic collapse of the forgetting process due to too fast divergence [20]. To address this shortcoming, NPO make the preference optimization framework

applicable to LLM unlearning task. This method extends the DPO to treat forget data as negative sample. The loss function of NPO is defined as follows:

$$\mathcal{L}_{NPO}\left(\pi_\theta, \pi_{\text{ref}}\right) = \mathbb{E}_{(x,y)\sim\mathcal{D}_{\text{f}}} \left[ -\frac{2}{\beta} \log \sigma \left( -\beta \log \frac{\pi_\theta\left(y \mid x\right)}{\pi_{\text{ref}}\left(y \mid x\right)} \right) \right] \tag{5}$$

According to Equation (4) and Equation (5), the gradient of GA and NPO can be expressed as:

$$\nabla_\theta \mathcal{L}_{\text{GA}} = \mathbb{E}_{\mathcal{D}_{\text{f}}} \left[ \nabla_\theta \log \pi_\theta(y \mid x) \right], \tag{6}$$

$$\nabla_\theta \mathcal{L}_{\text{NPO},\beta} = \mathbb{E}_{\mathcal{D}_{\text{f}}} \left[ W_\theta(x,y)\nabla_\theta \log \pi_\theta(y \mid x) \right], \tag{7}$$

where $W_\theta(x,y) = \frac{2\pi_\theta^\beta(y|x)}{\pi_\theta^\beta(y|x)+\pi_{\text{ref}}^\beta(y|x)}$ can be interpreted as an adaptive smoothing weight. When the sample $(x,y)$ is already unlearned, we have $W_\theta(x,y) \ll 1$, because of $\pi_\theta\left(y \mid x\right) \ll \pi_{\text{ref}}\left(y \mid x\right)$, so that $\|\nabla_\theta \mathcal{L}_{\text{NPO},\beta}\|_2 \ll \|\nabla_\theta \mathcal{L}_{\text{GA}}\|_2$. This indicates that NPO diverges more slowly than GA and thus has higher stability.

**Rigid Reward Setting.** In this paper, rigid reward setting is defined as the use of a fixed and inflexible reward mechanism to guide the model's learning direction in the unlearning process of LLM. There are two main types: reference-based reward and reference-free reward. The reference-based reward is based on an explicit reference model, which represents the initial state of the model before unlearning. By comparing the state of the current model with the reference model, the unlearning degree of the current model can be measured. While the reference-free reward adopts a constant offset $\gamma$ to replace the role of the reference model. In view of the many limitations exposed by rigid reward setting in practical applications (see Section B for details), we have carefully constructed an LLM unlearning framework based on elastic reward setting.

## 3 ERU: Elastic Robust Unlearning

In this section, we will explore in depth the proposed Elastic Robust Unlearning (ERU) framework. As shown in Figure 1, ERU is divided into two main parts. Firstly, we design an unlearning strategy based on elastic reward setting to overcome the limitations of rigid reward setting. Secondly, we incorporate the refusal feature ablation into the unlearing process and transform the unlearing robustness problem (see Section C for details) into a max-minimum optimization problem for solution.

### 3.1 Derive Elastic Reward Setting Objective

**Revisit the Rigid Reward Setting.** Start by revisiting the rigid reward setting. According to Equation (3), DPO gives reference-based reward:

$$R_{DPO}^{ref-based} = \beta \log \frac{\pi_\theta\left(y \mid x\right)}{\pi_{\text{ref}}\left(y \mid x\right)} \tag{8}$$

through comparison with the reference model. Assume $U\left(y \mid x\right)$ represents a uniform distribution over the vocabulary for a given input $x$, and replaces $\pi_{\text{ref}}\left(y \mid x\right)$ in the DPO loss function with it. In this way, DPO loss function can be simplified to:

$$\mathcal{L}_{DPO}^{ref-free}\left(\pi_\theta, U\right) = -\mathbb{E}_{(x,y_w,y_l)\sim\mathcal{D}} \left[\log \sigma \left(\beta \left(\log \pi_\theta\left(y_w \mid x\right) - \log \pi_\theta\left(y_l \mid x\right)\right) - \gamma\right)\right], \tag{9}$$

where $\gamma = \beta\left(\log U\left(y_w \mid x\right) - \log U\left(y_l \mid x\right)\right)$ is a constant called target reward margin, proposed by [33], and $\beta$ is a constant that controls the scaling of the reward difference. Consequently, this achieves a transition from reference-based reward to reference-free reward.

Equation (9) can be viewed as a special case of DPO where the reference model is uniformly distributed and $\gamma \geq 0$. This ensures that the reward difference between the preferred response and the less preferred response is controlled only by the policy model $\pi_\theta$, while making the following impossible:

$$\log \pi_{\text{ref}}\left(y_w \mid x\right) - \log \pi_{\text{ref}}\left(y_l \mid x\right) < 0, \tag{10}$$

thereby addressing the issue that the reference model $\pi_{\text{ref}}$ may incorrectly distinguish between the preferred and less preferred responses. These two rewards still fall into the category of rigid reward setting.

Similar to the above derivation, for NPO, because the reference-based reward is the same as for DPO, it is also possible to achieve the transition from reference-based to reference-free reward. Based on the design idea that NPO treats the forget data as negative examples in DPO, the NPO loss function with reference-free reward can be expressed as:

$$\mathcal{L}_{NPO}^{ref-free}\left(\pi_\theta, U\right) = \mathbb{E}_{(x,y)\sim\mathcal{D}_f}\left[-\frac{2}{\beta}\log\sigma\left(-\beta\log\pi_\theta\left(y\mid x\right)-\gamma'\right)\right] \tag{11}$$

where $\gamma' \geq 0$ is the variation of target reward margin and at this point the form of $\gamma'$ changes to:

$$\gamma' = \beta\left(-\log U\left(y\mid x\right)\right). \tag{12}$$

**Elastic Reward Setting.** As discussed in Section B, under rigid reward setting, relying solely on the reference model can lead to early-stage gradient weight smoothing ineffective, while full reliance on a constant offset may lose crucial instance distinctions. Therefore, we introduce the elastic reward setting to combine the advantages of both and compensate for their shortcomings.

First, we redefine $\pi_{\text{ref}}$ as a joint reference model $\hat{\pi}_{\text{ref}}$ in the following form:

$$\hat{\pi}_{\text{ref}}\left(y\mid x\right) = U(y\mid x)\left(\frac{\pi_\theta(y\mid x)}{\pi_{\text{ref}}\left(y\mid x\right)}\right)^\alpha, \tag{13}$$

where $\alpha$ is a hyperparameter controlling the influence of the policy model on the reference model. When $\alpha = 0$, $\hat{\pi}_{\text{ref}}$ is reduced to a uniform distribution in the reference-free reward. When $\alpha = 1$, it takes into account the ratio between the policy model and the reference model as in the NPO.

By substituting $\hat{\pi}_{\text{ref}}$ into Equation (5), we get the following new objective (See Section D.1 for detailed derivation.):

$$
\begin{aligned}
&\mathcal{L}^{new}\left(\pi_\theta, \hat{\pi}_{\text{ref}}, U\right) \\
&= \mathbb{E}_{(x,y)\sim\mathcal{D}_f}\left[-\frac{2}{\beta}\log\sigma\left(-\beta\log\frac{\pi_\theta\left(y\mid x\right)}{\hat{\pi}_{\text{ref}}\left(y\mid x\right)}\right)\right] \\
&= \mathbb{E}_{(x,y)\sim\mathcal{D}_f}\left[-\frac{2}{\beta}\log\sigma\left(-\beta\log\pi_\theta\left(y\mid x\right)-M\right)\right]
\end{aligned}
\tag{14}
$$

where $M$ is called the elastic reward margin and the formula is as follows:

$$M = \left[\gamma' - \alpha\left(\frac{\beta\log\left(\frac{\pi_\theta(y\mid x)}{\pi_{\text{ref}}(y\mid x)}\right) - \mu^*}{\sigma^*}\right)\right]. \tag{15}$$

The $\gamma' = \beta\left(-\log U\left(y\mid x\right)\right)$ is the same reward margin parameter as Equation (12), which is essentially a constant offset. Term $\beta\log\left(\pi_\theta(y\mid x)/\pi_{\text{ref}}(y\mid x)\right)$ measures the divergence in response pairs between the policy model $\pi_\theta$ and the reference model $\pi_{\text{ref}}$, effectively capturing the instance-specific discrepancies. In order to avoid $\beta\log\left(\pi_\theta(y\mid x)/\pi_{\text{ref}}(y\mid x)\right)$ dominating training due to scale variations, we perform z-score normalization[34] on it, where $\mu^*$ and $\sigma^*$ are its mean and standard deviation calculated over the training dataset. In addition, as a fixed reference during the optimization, the reference model should maintain gradients unchanged throughout the training process. For this purpose, we use the stop gradient operation to ensure, denoted as $\text{rg}\left[\cdot\right]$. Combined with the above considerations and length normalization, the final loss becomes:

$$\mathcal{L}_{EU}(\pi_\theta, \hat{\pi}_{\text{ref}}, U) = \mathbb{E}_{(x,y)\in\mathcal{D}_f}\left[-\frac{2}{\beta}\log\sigma\left(u(x,y)-\text{rg}\left[M\right]\right)\right], \tag{16}$$

where $u(x,y) = -\frac{\beta}{|y|}\log\pi_\theta(y\mid x)$. Equation (16), which called Elastic Unlearning (EU), can effectively slow down the rapid drop of gradient weights in the early stage caused by reference-based reward (See Section D.2 for theoretical analysis), and avoid complete reliance on reference-free reward. By incorporating an elastic reward margin $M$, it considers the balanced influence between the policy model and the reference model. The following sections will focus on how to enhance the unlearning robustness of EU.

## 3.2 Refusal Feature Ablation for Elastic Unlearning

Recent studies [35, 36] have shown that introducing adversarial training in the LLM unlearning process can significantly improve the robustness of the unlearning method. However, existing methods often have high computational costs, which is contrary to the original intention of LLM unlearning. Inspired by recent research on refusal feature (RF) [37, 26], we enhance the unlearning robustness of Elastic Unlearning through refusal feature ablation (RFA) to construct Elastic Robust Unlearning.

**Refusal Feature Ablation.** Research [26] shows that a key mechanism of adversarial attacks is to eliminate refusal features, making it difficult for the model to identify the harmfulness of the input. RFA simulates this attack behavior by directly eliminating the refusal features in the hidden representation. Following [37], given a collection of harmful prompts of tokens $x = [t_1, ..., t_j] \in \mathcal{D}_{\text{harmful}}$ and another set of harmless prompts $x \in \mathcal{D}_{\text{harmless}}$, the refusal feature of each layer $l \in L$ in LLM $\pi_\theta$ is a one-dimensional feature linearly encoded in the residual stream, which is obtained by calculating the difference between the model's mean last-token residual stream activations $\mathbf{h}^{(l)}(x)$ when running on harmful and harmless inputs:

$$\mathbf{r}_{\text{HH}}^{(l)} = \frac{1}{|\mathcal{D}_{\text{harmful}}|} \sum_{x \in \mathcal{D}_{\text{hamful}}} \mathbf{h}^{(l)}(x) - \frac{1}{|\mathcal{D}_{\text{harmless}}|} \sum_{x \in \mathcal{D}_{\text{harmless}}} \mathbf{h}^{(l)}(x) \tag{17}$$

where $\mathcal{D}_{\text{harmful}}$ and $\mathcal{D}_{\text{harmless}}$ by sampling 500 instructions from the AdvBench [38] and the Alpaca [39] datasets respectively. Furthermore, RFA is defined as an inference-time intervention that sets the refusal feature at each layer as its average activation on harmless prompts:

$$\mathbf{h}'^{(l)}(x) \leftarrow \mathbf{h}^{(l)}(x) - \hat{\mathbf{r}}\hat{\mathbf{r}}^T \mathbf{h}^{(l)}(x) + \frac{1}{|\mathcal{D}_{\text{harmless}}|} \sum_{x \in \mathcal{D}_{\text{harmless}}} \hat{\mathbf{r}}\hat{\mathbf{r}}^T \mathbf{h}^{(l)}(x) \tag{18}$$

where $\hat{\mathbf{r}} = \frac{\mathbf{r}_{\text{HH}}^{(l)}}{\left\| \mathbf{r}_{\text{HH}}^{(l)} \right\|}$ is unit vector encoding the refusal feature direction, and $\mathbf{h}^{(l)}(x) - \hat{\mathbf{r}}\hat{\mathbf{r}}^T \mathbf{h}^{(l)}(x)$ is the projection that resets the value to zero along the refusal direction, and the last item is set to patch the the refusal feature. In order to improve the unlearning robustness of the model, we next integrate RFA into the training process of EU.

**Elastic Robust Unlearning (ERU).** We describe adversarial training applied to LLM Unlearning as a max-min optimization problem. This problem consists of an inner maximization process and an outer minimization process. The goal of the internal maximization process is to continuously increase the prediction probability of forgotten knowledge as much as possible through adversarial queries that may effectively bypass the model's limitations. The objective of outer minimization is to suppress the reemergence of forgotten knowledge, making its prediction probability as low as possible. In standard adversarial training with an $L_p$-norm constraint of $\epsilon$, unlearned models will be trained to be robust to adversarial queries with perturbation $\delta$ in the input space:

$$\min_\theta \mathbb{E}_{(x,y) \sim \mathcal{D}_{\text{f}}} \max_\delta \mathcal{L}\left(\pi_\theta\left(x + \delta, y\right)\right) \quad \text{s.t.} \ \|\delta\|_p \leq \epsilon. \tag{19}$$

where $\mathcal{L}$ is the unlearning loss function. To avoid generating adversarial samples through complex attack algorithms as in the traditional adversarial training method, we take RFA as an effective attack simulation during adversarial training.

We treat LLM with parameter $\theta$ as a combination of two functions, where $\mathbf{H}(x) = \left\{ \mathbf{h}^{(l)}(t_j) \right\}_{l=1}^{L}$ is the hidden representations that focus on the residual stream of the last token $t_j$, and $g_\theta$ maps those hidden representations to output a probability distribution for sampling i.e., $\pi_\theta\left(y \mid g_\theta \circ \mathbf{H}(x)\right)$. Unlike general adversarial training, refusal feature adversarial training (RFAT) is performed by simulating the perturbation of RFA:

$$\underbrace{\pi_\theta\left(g_\theta \circ \left(\mathbf{H}\left(x + \delta\right)\right), y\right)}_{\text{Input-space perturbation}} \Rightarrow \underbrace{\pi_\theta\left(g_\theta \circ \left(\mathbf{H}\left(x\right) - \mathbf{R}_{\text{HH}}\right), y\right)}_{\text{Perturbations simulated by RFA}} \tag{20}$$

where $\mathbf{R}_{\text{HH}} = \left\{ \mathbf{r}_{\text{HH}}^{(l)} \right\}_{l=1}^{L}$ is the layerwise refusal feature that it needs to be dynamically updated every certain training steps, and $\mathbf{H}(x) - \mathbf{R}_{\text{HH}}$ denotes the removal of refusal features across model

Table 1: Overview of the unlearning effectiveness of various PO-based unlearning methods using the LLaMA-2-7B-Chat across four unlearning benchmarks. The best results are highlighted in **bold**, and the next best results are highlighted by underlining "__".

| Method | RWKU | | | MUSE-News | | | TOFU | | WMDP | |
|---|---|---|---|---|---|---|---|---|---|---|
| | Forget Set | | | VerbMem ($\mathcal{D}_f \downarrow$) | KnowMem ($\mathcal{D}_f \downarrow$) | PrivLeak ($\in [-5\%, 5\%]$) | Forget05-FQ↑ | Forget10-FQ↑ | AccBio↓ | AccCyber↓ |
| | FB↓ | QA↓ | AA↓ | | | | | | | |
| Original | 51.9 | 46.8 | 57.5 | 58.3 | 63.7 | -99.8 | 3.2e-16 | 2.1e-19 | 63.2 | 42.8 |
| DPO | 38.9 | 40.7 | 41.5 | 33.2 | 37.2 | 109.6 | 1.2e-4 | 3.5e-7 | 28.9 | 33.5 |
| IDK | 40.5 | 40.6 | 45.4 | 35.6 | 39.1 | 104.3 | 4e-5 | 5e-8 | 29.3 | 34.2 |
| GA | 44.5 | 39.6 | 47.3 | **0.0** | **0.0** | 20.8 | 0.05 | 8.1e-10 | 37.4 | 30.1 |
| GradDiff | 46.4 | 42.2 | 48.6 | 25.9 | 31.0 | 105.3 | 0.09 | 7.9e-3 | 38.6 | 33.5 |
| GA$_{KLR}$ | 46.8 | 41.4 | 44.3 | 27.4 | 58.6 | -51.6 | 0.11 | 3.4e-5 | 37.9 | 33.2 |
| NPO | 33.6 | 31.3 | 32.8 | 10.8 | 13.4 | 30.4 | 0.66 | 0.19 | 29.6 | 32.7 |
| NPO$_{GDR}$ | 34.8 | 34.7 | 38.1 | 13.2 | 48.6 | 101.3 | 0.44 | 0.24 | 31.8 | 33.0 |
| NPO$_{KLR}$ | 37.6 | 34.5 | 38.5 | 16.6 | 38.6 | -56.7 | 0.43 | 0.17 | 32.4 | 32.9 |
| SimNPO | 34.2 | 31.8 | 37.5 | 12.6 | 11.3 | 14.9 | **0.97** | 0.45 | 28.6 | 29.8 |
| AdvNPO | 35.9 | 33.2 | **25.2** | 13.7 | 12.8 | 24.6 | 0.63 | 0.26 | 29.9 | 33.2 |
| ERU | **29.2** | **27.1** | 25.5 | 10.4 | 9.2 | **12.3** | 0.73 | **0.48** | **24.8** | **28.4** |
| ERU$_{GDR}$ | 29.8 | 28.6 | 25.9 | 11.4 | 10.8 | 14.1 | 0.69 | 0.45 | 25.7 | 29.4 |
| ERU$_{KLR}$ | 31.4 | 27.9 | 27.5 | 11.6 | 11.2 | 15.3 | 0.67 | 0.42 | 27.5 | 30.2 |

layers. In an actual adversarial environment, attackers may not always succeed in eliminating refusal features. Following [37], we perform RFA with probability $p$ to approximate the different degrees of adversarial perturbations encountered by the model during the training process, thus better simulating the adversarial attack scenarios in the real world. Therefore, RFAT for EU to construct ERU can be expressed as:

$$\min_{\theta} \mathbb{E}_{(x,y)\sim\mathcal{D}_f} \max_{\delta} \mathcal{L}\left(p\pi_\theta\left(g_\theta \circ \left(\mathbf{H}(x) - \mathbf{R}_{HH}\right), y\right) + (1-p)\pi_\theta\left(g_\theta \circ \left(\mathbf{H}(x)\right), y\right)\right) \tag{21}$$

Following [26], we applied RFA on the residual stream activations over the last 75% layer of each model to obtain the most stable fine-tuning results (see Section E.2.1 for detailed settings).

## 4 Experiments

### 4.1 Experimental Setup

**Datasets and Models.** Our experiments cover unlearning tasks across four benchmarks: RWKU [27], MUSE [28], TOFU [29], and WMDP [30]. Among them, RWKU is a real-world knowledge unlearning benchmark specifically for LLM unlearning, while WMDP is designed to prevent LLMs from generating harmful content in fields such as biology, cybersecurity, and chemistry. For TOFU, we explore two unlearning scenarios, termed Forget05 and Forget10, representing forget set sizes of 5% and 10%, respectively. For the MUSE dataset, we primarily focus on the unlearning scenario of news articles. The experiments are conducted based on the LLaMA-2-7B-Chat [3] and LLaMA-3-8B-Instruct [31]. More details are provided in Section E.1.

**Evaluation Metrics.** Despite differences in the evaluation metrics of the aforementioned tasks, the evaluation framework can be summarized into the following three categories:**(1) Unlearning Effectiveness**, which measures the ability of unlearning methods to successfully remove the influence or behavior of specific data from the model. **(2) Utility Preservation**, which evaluates the performance of the unlearned model in executing standard tasks. **(3) Unlearning Robustness**, referring to the model's ability to resist relearning or recovering the unlearned knowledge. The summary of the metrics on different unlearning benchmarks is shown in Table 6, with detailed information available in Section E.1 and Section E.2.

**Baseline LLM Unlearning Methods.** Researchers have proposed various efficient and practical unlearning methods. To evaluate the performance of our proposed ERU framework, we select the most representative preference optimization-based unlearning methods as baselines. First, we refer to the model without any unlearning as **Original**, which reflects the initial performance of the model across various tasks. In the selection of baseline methods, in addition to the **GA**, **NPO**, and **DPO**

mentioned in Section 2.3, we also included **GradDiff** (a retention-regularized variant of GA), the rejection-based unlearning method **IDK**, **SimNPO** [22] which relies on rigid reward setting, and the **AdvNPO** [35] which enhances unlearning robustness through latent-space adversarial training. Furthermore, we define new baselines($\text{GA}_{\text{KLR}}$, $\text{NPO}_{\text{GDR}}$, and $\text{NPO}_{\text{KLR}}$) by integrating the GA and NPO methods with two regularizers. More details are presented in Section E.3.

## 4.2 Results of Unlearning Effectiveness Evaluation

The unlearning effectiveness reflects the extent to which unlearning methods remove specific knowledge from LLMs, so we aim for this metric to be as low as possible. Table 1 shows the experimental results of various unlearning methods based on LLaMA-2-7B-Chat across four benchmarks. As can be seen from the table, existing methods generally underperform compared to our ERU. Specifically, ERU achieves a 16.3% higher average metric than the second-best method (NPO) on the RWKU benchmark, and outperforms the second-best method by 6% on the WMDP benchmark. This trend remains evident in the TOFU benchmark evaluation. It should be noted that the GA method has achieved the best performance in some metrics of the MUSE benchmark, indicating that it does not generate any text related to the forget set. This extreme performance comes at the cost of significant losses of model utility (as shown in Table 2), making the unlearned models generated by GA almost unusable. The PrivLeak metric in MUSE further demonstrates that existing methods generally suffer from over-unlearning or under-unlearning, while ERU performs best in this respect. In summary, ERU performs well in unlearning effectiveness, significantly superior to baseline methods. See Table 7 in Section F.1 for more experimental results.

## 4.3 Impact of Refusal Feature Adversarial Training

To evaluate the impact of refusal feature adversarial training on unlearning methods, we employ relearning attacks and various adversarial attacks to test their robustness (see Section E.2 for attack details). Specifically, we evaluate the unlearing robustness of different unlearning methods by comparing the changes in unlearning performance on the WMDP-Bio benchmark before and after the unlearned models are attacked.

For relearning attacks, given that the malicious user can recover partially forgotten knowledge in unlearned models by fine-tuning irrelevant information, we fine-tune two types of data sets: (1) the retain set; (2) WikiText, a collection of documents on Wikipedia that overlap least with dangerous knowledge. As shown in Figure 2, the experimental results show that fine-tuning with only 10 samples of the retain set can significantly restore the dangerous capability of most methods on WMDP-Bio, among which GA, DPO, NPO, SimNPO methods can restore 72.9%, 80.2%, 69.6% and 34.3% unlearning performance, respectively. Further fine-tuning with 1000 samples from the retain set, the dangerous capabilities of all methods except the AdvNPO were almost fully restored. In contrast, ERU recovers merely 8.1% of its unlearning effectiveness when fine-tuned with 10 samples, and retains 83% of its unlearning effectiveness even after fine-tuning with 1,000 samples. More detailed experimental results can be found in Section F.6.

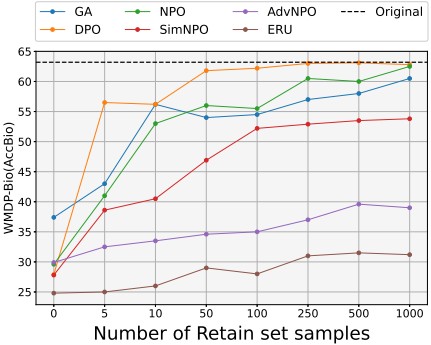

Figure 2: Knowledge recovery of different unlearing methods on the WMDP-Bio after fine-tuning with different numbers of retain set samples.

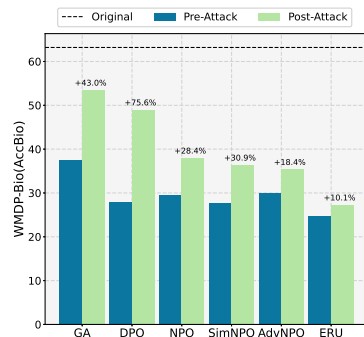

Figure 3: Comparison of knowledge recovery of different unlearing methods under Enhanced GCG.

For the robustness evaluation of adversarial attacks, we test the unlearned model with nine different types of attacks to induce it to generate forgotten knowledge. Figure 3 shows the degree of knowledge recovery of different unlearned models under Enhanced GCG attack, which based on an optimized set of adversarial input prefixes. Experimental results show that on the WMDP-Bio benchmark, the dangerous knowledge of most unlearned models is recovered by more than 50%, among which the DPO method has the most significant improvement, reaching 75.6%. In baseline methods, ANPO performed best, but its dangerous knowledge recovery rate was still 18.4%, compared to only 10.1% for ERU. This comparison fully confirms that the ERU framework has significant advantages in terms of robustness. More detailed experimental results are shown in Section F.7.

Table 2: Utility preservation of various PO-based unlearning methods on LLaMA-2-7B-Chat.

| Method | RWKU (Utility Set) | | | | MUSE-News | MMLU | TOFU-Forget05 | | TOFU-Forget10 | |
| | Rea↑ | Tru↑ | Fac↑ | Flu↑ | KnowMem($\mathcal{D}_r$ ↑) | Accuracy↑ | Probability↑ | ROUGE↑ | Probability↑ | ROUGE↑ |
|---|---|---|---|---|---|---|---|---|---|---|
| Original | 26.9 | 30.4 | 41.5 | 704.2 | 55.2 | 58.5 | 0.99 | 0.98 | 0.99 | 0.98 |
| DPO | 26.4 | 25.2 | 32.4 | 710.6 | 32.8 | 46.8 | 0.74 | 0.53 | 0.76 | 0.54 |
| IDK | **26.8** | 27.9 | 36.7 | **712.5** | 37.3 | 50.2 | **0.76** | 0.55 | 0.78 | 0.55 |
| GA | 25.8 | 30.7 | 40.2 | 707.6 | 0.00 | 48.3 | 0.00 | 0.00 | 0.00 | 0.00 |
| GradDiff | 24.8 | 30.4 | 41.1 | 707.5 | 27.3 | 51.2 | 0.49 | 0.42 | 0.57 | 0.48 |
| GA$_{KLR}$ | 26.2 | 29.8 | 40.6 | 708.3 | 41.8 | 51.8 | 0.48 | 0.44 | 0.53 | 0.49 |
| NPO | 26.2 | 30.5 | 41.1 | 694.6 | 27.5 | 47.6 | 0.51 | 0.47 | 0.46 | 0.44 |
| NPO$_{GDR}$ | 26.5 | 30.4 | 40.8 | 705.2 | 40.5 | 51.7 | 0.56 | 0.55 | 0.65 | 0.53 |
| NPO$_{KLR}$ | 26.3 | **31.2** | 40.9 | 703.8 | 46.4 | 50.5 | 0.56 | 0.54 | 0.71 | 0.55 |
| SimNPO | 26.3 | 29.4 | 40.5 | 691.3 | 43.5 | 50.2 | 0.56 | 0.54 | 0.72 | 0.53 |
| AdvNPO | 24.3 | 26.5 | 39.8 | 672.8 | 24.3 | 41.2 | 0.48 | 0.46 | 0.46 | 0.45 |
| ERU | 26.2 | 30.5 | 40.5 | 708.8 | 43.2 | 50.6 | 0.59 | 0.56 | 0.74 | 0.53 |
| ERU$_{GDR}$ | 26.1 | 30.7 | **41.2** | 707.5 | **47.2** | 52.1 | 0.72 | **0.57** | **0.79** | **0.56** |
| ERU$_{KLR}$ | 26.6 | 30.8 | 40.9 | 708.6 | 44.2 | **53.4** | 0.74 | **0.57** | 0.78 | 0.55 |

## 4.4 Utility Preservation Analysis

An effective unlearning method should keep the model as available as possible. The results in Table 2 show that all unlearning methods cause some degree of utility damage to the LLMs, with GA and AdvNPO leading to particularly significant model utility loss, the former rendering the unlearned model almost unusable. However, by introducing the utility regularization term, the utility preservation of these methods is significantly improved, though still falling short of the original model. In contrast, ERU maintains high utility preservation even without the introduction of regularization terms, while the incorporation of regularization terms resulted in a better utility preservation of ERU.

Table 3: Comparison of training time of different LLM unlearning methods.

| Method | NPO | SimNPO | AdvNPO | EU | ERU |
|---|---|---|---|---|---|
| Time (min) | 156 | 92 | 539 | 125 | 243 |

## 4.5 Efficiency Analysis of ERU

Compared with the existing unlearning methods for robustness enhancement based on adversarial training, ERU has similar or better robustness performance, but is more efficient. This is because ERU does not need to perform complex gradient calculations and multiple iterative optimizations through dynamic simulation of attack algorithms (such as PGD [40]) to find adversarial samples. Instead, it directly performs ablation operations on the refusal features, significantly reducing additional forward and backward passes and greatly lowering the computational cost. Table 3 presents the comparison of training time consumption of different unlearning methods in the same training environment in this paper. It can be clearly seen from the table that the AdvNPO method significantly increases the time cost by enhancing the unlearning robustness of NPO through latent-space adversarial training. In contrast, the increase in time cost of the ERU method has been significantly reduced. See Section E.5 for further details.

## 5 Conclusion

In this paper, we propose the Elastic Robust Unlearning (ERU), a novel framework to enhance the unlearning capabilities of LLMs. Our framework addresses two critical limitations of PO-based unlearning methods: the rigid reward setting and the lack of unlearning robustness. By proposing the elastic reward setting, we achieve a more flexible balance between reference-based reward and reference-free reward, which significantly improved the unlearning effectiveness while maintaining model utility. Furthermore, the incorporation of refusal feature ablation into the unlearning process significantly boosts the robustness of the unlearned models. Our extensive experiments across multiple benchmarks and LLMs have demonstrated the superiority of ERU in terms of unlearning effectiveness, utility preservation, and unlearning robustness.

## Acknowledgments and Disclosure of Funding

This research is supported by the Natural Science Foundation of China (Nos. 62032024, 62372459), the Natural Science Foundation of Hunan Province, China (2024JK2015), the Special Foundation for Distinguished Young Scientists of Changsha (kq2209003), the Foundation of State Key Laboratory of High Performance Computing, National University of Defense Technology (202401-13).

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

# Appendix

## A   Related Work

**LLM Unlearning.**   The goal of unlearning is to make a trained model perform as if it were untrained on specific datasets by removing specific knowledge [41]. In the field of language models, research on unlearning has expanded into themes such as fairness, privacy, safety or hallucinations [42, 43]. Due to the need for large-scale computing costs, traditional unlearning methods are difficult to directly apply to LLMs, so a growing number of studies began to focus on the LLM unlearning [18, 44, 45, 18, 46, 29]. Yao et al. [18] were the first to define the setup and objectives of LLM unlearning. Current LLM unlearning methods are mainly divided into two categories: in-context learning-based unlearning and model optimization-based unlearning. The former, such as labeled demonstrations or post-processing filters, cannot completely remove specific knowledge from the model weights[47, 48, 46]. The latter minimizes correct predictions of forget objects through methods such as Gradient Ascent (GA) [18]. In recent studies, Negative Preference Optimization (NPO) [20] has been proposed as a promising approach. It treats the unlearning task as a variant of Direct Preference Optimization (DPO) [19], aiming to solve the above challenges more effectively. Our research is committed to exploring the operational mechanism of PO-based LLM unlearning in depth, identifying the main limitations it currently faces, and making practical recommendations for improvement based on these findings.

**Preference Optimization.**   The motivation for preference optimization is similar to that of LLM unlearning, which is to align LLM with human values. Viewing from the lens of preference optimization, we can better understand its connection with LLM unlearning. The core idea of preference optimization is derived from the reinforcement learning from human feedback (RLHF) [49, 50], of which direct preference optimization (DPO) is a typical example. As an offline improvement of online preference optimization algorithms, DPO removes the need for an explicit reward model, thereby spurring the development of various reward-free offline preference objectives, including RRHF [51], SLic-HF [52], IPO [53], KTO [54], ORPO [55], and SimPO [33]. Recently, inspired by DPO, Zhang et al. treated LLM unlearning as a special case of DPO without positive samples, proposing the NPO [20]. This not only establishes the theoretical connection between preference optimization and LLM unlearning, but also provides a new research perspective in this field. Following this research trajectory, we focus on improving preference-based LLM unlearning methods. Different from the existing approaches that use rigid reward setting to assign reward values to different outputs during training, we propose the elastic reward setting that dynamically balances reference-based reward and reference-free reward, effectively combine the advantages of both and avoid their limitations.

**Unlearning Robustness.**   Despite the effectiveness of LLM unlearning in removing harmful information from model weights, the resulting unlearned models tend to be fragile. Studies have shown that unlearned models are easily induced by adversarial prompts to regenerate deleted knowledge [35, 27]. Lynch et al. [56] further found that knowledge can be extracted from both the unlearned model with high efficiency by analyzing the internal representation of the model. Furthermore, Łucki et al. [57] proposed a new method to extract potentially dangerous knowledge from unlearned models without updating the model weights. Hu et al. [58] show that access to only a small number of possibly loosely related datasets can "wake up" the unlearned model and reverse the unlearning effect. These findings suggest that current LLM unlearning techniques require further improvement to enhance their robustness. Recent work [37, 26] on refusal feature prove that refusal feature ablation can be used to approximate the worst-case perturbation in adversarial training, and we aim to extend it to the aspect of unlearing robustness to construct a robust LLM unlearing framework.

## B   Limitations of Rigid Reward Setting

Recent research [22] indicates that this reference-based reward setting may result in poor gradient weight smoothing. The reward formula in NPO is defined as $\beta \log(\pi_\theta (y \mid x)/\pi_{\mathrm{ref}} (y \mid x))$. Since the unlearned model is initialized identically to the reference model at the beginning of training, the

weight gradient is approximately equal to 1 at this point (Figure 4a), i.e.:

$$W_\theta^{init}(x, y) = \frac{2\pi_\theta^\beta(y \mid x)}{\pi_\theta^\beta(y \mid x) + \pi_{\text{ref}}^\beta(y \mid x)} \approx 1. \tag{22}$$

This suggests that NPO behaves like GA in the early stages of unlearning, which may lead to over-unlearning even if the weight is reduced in the subsequent optimization process. Especially in the early stages of training, NPO tends to lead to a greater reduction in utility (Figure 4b).

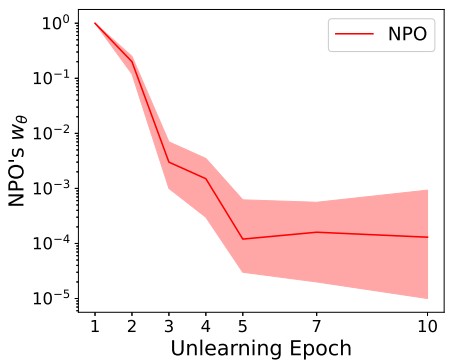 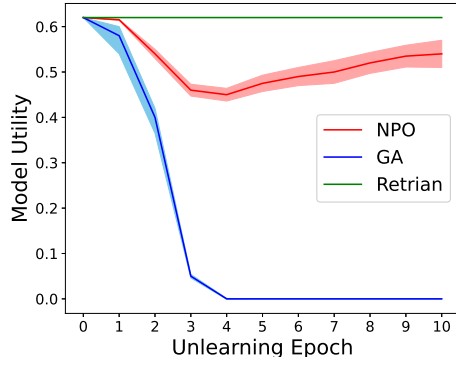

(a) Trajectory of $W_\theta$ for NPO over unlearning epochs.

(b) Model utility of NPO and GA across epochs.

Figure 4: Experimental phenomena of NPO and GA on forget05 dataset of TOFU.
For reference-free reward, a constant offset $\gamma$ is typically employed to substitute the role of the reference model. However, the limitation of this approach is that $\gamma$ remains unchanged across all training samples, failing to effectively capture the variability inherent between different data instances [59]. Completely discarding the reference model may lead to a decline in model performance.

## C   Limitations of Unlearning Robustness

While unlearning techniques are effective at removing harmful knowledge from LLMs, their robustness may be weakened by methods similar to those used for safe training. For example, fine-tuning with just 10 unrelated samples can significantly reduce the unlearning effect and restore the original performance of the model. In addition, malicious users may bypass restrictions with improved adversarial prompts to regain access to unlearned knowledge.

Taking the GCG attack [38] as an example, although the unlearned model can effectively defend against the GCG attack, the attack effect can be restored by small adjustments to the GCG loss function. Figure 5 shows the knowledge recovery effects of six methods on unlearned models, and the results show that all methods can invalidate unlearning to a certain extent. Current criteria for evaluating the effects of unlearning are often based on non-adversarial scenarios and lack consideration of situations where malicious users induce models to regenerate deleted knowledge. To this end, it is necessary to study new unlearning methods in the adversarial context and

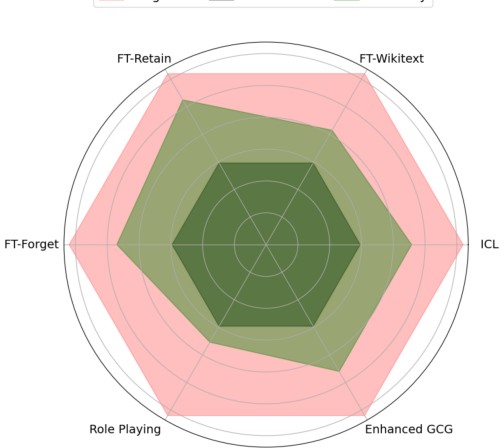

Figure 5: The unlearning robustness of NPO against knowledge recovery attacks and adversarial attacks by evaluating the accuracy on the WMDP-Bio benchmark. See Section E.2.3 for details on adversarial queries such as In-context Learning (ICL), Role Playing and Enhanced GCG.

incorporate robustness evaluation into the comprehensive evaluation system of unlearned models.

# D  Theoretical Analysis

## D.1  Derivation of ERU Loss.

The known joint reference model $\hat{\pi}_{\text{ref}}$ is defined as:

$$\hat{\pi}_{\text{ref}}(y \mid x) = U(y \mid x)\left(\frac{\pi_\theta(y \mid x)}{\pi_{\text{ref}}(y \mid x)}\right)^\alpha, \tag{23}$$

where $\alpha$ is the hyperparameter of the influence of the reference model. Substituting $\hat{\pi}_{\text{ref}}$ into the loss function of NPO, a new objective function is obtained:

$$\mathcal{L}^{new}(\pi_\theta, \hat{\pi}_{\text{ref}}, U) = \mathbb{E}_{(x,y)\sim\mathcal{D}_{\text{f}}}\left[-\frac{2}{\beta}\log\sigma\left(-\beta\log\frac{\pi_\theta(y \mid x)}{\hat{\pi}_{\text{ref}}(y \mid x)}\right)\right]. \tag{24}$$

Expanding $\hat{\pi}_{\text{ref}}$ further, we have:

$$
\begin{aligned}
&\mathcal{L}^{new}(\pi_\theta, \hat{\pi}_{\text{ref}}, U) \\
&= \mathbb{E}_{(x,y)\sim\mathcal{D}_{\text{f}}}\left[-\frac{2}{\beta}\log\sigma\left(-\beta\log\frac{\pi_\theta(y \mid x)}{\hat{\pi}_{\text{ref}}(y \mid x)}\right)\right] \\
&= \mathbb{E}_{(x,y)\sim\mathcal{D}_{\text{f}}}\left[-\frac{2}{\beta}\log\sigma\left(-\beta\log\frac{\pi_\theta(y \mid x)}{U(y \mid x)\left(\frac{\pi_\theta(y|x)}{\pi_{\text{ref}}(y|x)}\right)^\alpha}\right)\right] \\
&= \mathbb{E}_{(x,y)\sim\mathcal{D}_{\text{f}}}\left[-\frac{2}{\beta}\log\sigma\left(-\beta\log\pi_\theta(y \mid x) - \left[-\beta\log U(y \mid x) - \alpha\beta\left(\frac{\pi_\theta(y \mid x)}{\pi_{\text{ref}}(y \mid x)}\right)\right]\right)\right].
\end{aligned} \tag{25}
$$

By substituting Equation (12), we can get:

$$
\begin{aligned}
&\mathcal{L}^{new}(\pi_\theta, \hat{\pi}_{\text{ref}}, U) \\
&= \mathbb{E}_{(x,y)\sim\mathcal{D}_{\text{f}}}\left[-\frac{2}{\beta}\log\sigma\left(-\beta\log\pi_\theta(y \mid x) - \left[-\beta\log U(y \mid x) - \alpha\beta\left(\frac{\pi_\theta(y \mid x)}{\pi_{\text{ref}}(y \mid x)}\right)\right]\right)\right] \\
&= \mathbb{E}_{(x,y)\sim\mathcal{D}_{\text{f}}}\left[-\frac{2}{\beta}\log\sigma\left(-\beta\log\pi_\theta(y \mid x) - \left[\gamma' - \alpha\beta\left(\frac{\pi_\theta(y \mid x)}{\pi_{\text{ref}}(y \mid x)}\right)\right]\right)\right].
\end{aligned} \tag{26}
$$

In order to stabilize training and avoid dominant loss due to scale variations, apply Z-score normalization on Equation (26):

$$
\begin{aligned}
&\mathcal{L}^{new}(\pi_\theta, \hat{\pi}_{\text{ref}}, U) \\
&= \mathbb{E}_{(x,y)\sim\mathcal{D}_{\text{f}}}\left[-\frac{2}{\beta}\log\sigma\left(-\beta\log\pi_\theta(y \mid x) - \left[-\beta\log U(y \mid x) - \alpha\beta\left(\frac{\pi_\theta(y \mid x)}{\pi_{\text{ref}}(y \mid x)}\right)\right]\right)\right] \\
&= \mathbb{E}_{(x,y)\sim\mathcal{D}_{\text{f}}}\left[-\frac{2}{\beta}\log\sigma\left(-\beta\log\pi_\theta(y \mid x) - \left[\gamma' - \alpha\left(\frac{\beta\log\left(\frac{\pi_\theta(y|x)}{\pi_{\text{ref}}(y|x)}\right) - \mu^*}{\sigma^*}\right)\right]\right)\right].
\end{aligned} \tag{27}
$$

## D.2  Analysis of ERU training stability.

Equation (16) is the final loss function of the ERU method proposed in this paper, and its form is as follows:

$$\mathcal{L}_{EU}(\pi_\theta, \hat{\pi}_{\text{ref}}, U) = \mathbb{E}_{(x,y)\in\mathcal{D}_{\text{f}}}\left[-\frac{2}{\beta}\log\sigma\left(u(x,y) - \text{rg}\,[M]\right)\right], \tag{28}$$

where

$$u(x,y) = -\frac{\beta}{|y|}\log\pi_\theta(y \mid x) \tag{29}$$

and $M$ is the elastic reward margin. Since $M$ uses the stop gradient operation $\text{rg}\,[\cdot]$, the gradient of $M$ is not updated during backpropagation, so $\text{rg}\,[M]$ is treated as a constant when calculating the gradient. We then analyze the gradient of the above loss function. First, to compute the gradient of $u(x,y)$, we have:

$$\nabla_\theta u(x,y) = -\frac{\beta}{|y|}\nabla_\theta\log\pi_\theta(y \mid x). \tag{30}$$

Define the variables:

$$z = u(x,y) - \text{rg}\,[M] = -\frac{\beta}{|y|} \log \pi_\theta(y \mid x) - \text{rg}\,[M]. \tag{31}$$

Therefore, Equation (28) becomes:

$$\mathbb{E}_{(x,y)\in\mathcal{D}_f} \left[ -\frac{2}{\beta} \log \sigma\,(z) \right], \tag{32}$$

Where $\sigma\,(\cdot)$ is the sigmoid function and its derivative is $\sigma'\,(\cdot) = \sigma\,(\cdot)\,(1 - \sigma\,(\cdot))$. Compute the gradient of the Equation (32) with respect to $u(x,y)$:

$$\nabla_{u(x,y)}\mathcal{L}_{EU} = \mathbb{E}_{(x,y)\in\mathcal{D}_f} \left[ -\frac{2}{\beta} \cdot \frac{1}{\sigma(z)} \cdot \sigma(z)(1 - \sigma(z)) \right] = \mathbb{E}_{(x,y)\in\mathcal{D}_f} \left[ -\frac{2}{\beta}(1 - \sigma(z)) \right]. \tag{33}$$

According to the chain rule, we have:

$$\nabla_\theta \mathcal{L}_{EU} = \mathbb{E}_{(x,y)\in\mathcal{D}_f} \left[ -\frac{2}{\beta}(1 - \sigma(z)) \cdot \nabla_\theta u(x,y) \right], \tag{34}$$

Substitute Equation (30) into Equation (34):

$$
\begin{aligned}
\nabla_\theta \mathcal{L}_{EU} &= \mathbb{E}_{(x,y)\in\mathcal{D}_f} \left[ -\frac{2}{\beta}(1 - \sigma(z)) \cdot -\frac{\beta}{|y|} \nabla_\theta \log \pi_\theta(y \mid x) \right] \\
&= \mathbb{E}_{(x,y)\in\mathcal{D}_f} \left[ 2 \cdot (1 - \sigma(z)) \cdot \frac{1}{|y|} \nabla_\theta \log \pi_\theta(y \mid x) \right],
\end{aligned} \tag{35}
$$

We further transform the Equation (35):

$$
\begin{aligned}
\nabla_\theta \mathcal{L}_{EU} &= \mathbb{E}_{(x,y)\in\mathcal{D}_f} \left[ 2 \cdot (1 - \sigma(z)) \cdot \frac{1}{|y|} \nabla_\theta \log \pi_\theta(y \mid x) \right] \\
&= \mathbb{E}_{(x,y)\in\mathcal{D}_f} \left[ 2 \cdot \left( 1 - \frac{\exp(z)}{\exp(z) + 1} \right) \cdot \frac{1}{|y|} \nabla_\theta \log \pi_\theta(y \mid x) \right] \\
&= \mathbb{E}_{(x,y)\in\mathcal{D}_f} \left[ \left( \frac{2}{\exp(z) + 1} \right) \cdot \frac{1}{|y|} \nabla_\theta \log \pi_\theta(y \mid x) \right] \\
&= \mathbb{E}_{(x,y)\in\mathcal{D}_f} \left[ \left( \frac{2 \cdot \exp(-z)}{\exp(-z) + 1} \right) \cdot \frac{1}{|y|} \nabla_\theta \log \pi_\theta(y \mid x) \right] \\
&= \mathbb{E}_{(x,y)\in\mathcal{D}_f} \left[ \left( \frac{2 \cdot \exp(\frac{\beta}{|y|} \log \pi_\theta(y \mid x) + \text{rg}\,[M])}{\exp(\frac{\beta}{|y|} \log \pi_\theta(y \mid x) + \text{rg}\,[M]) + 1} \right) \cdot \frac{1}{|y|} \nabla_\theta \log \pi_\theta(y \mid x) \right] \\
&= \mathbb{E}_{(x,y)\in\mathcal{D}_f} \left[ \left( \frac{2 \cdot \exp(\text{rg}\,[M]) \cdot [\pi_\theta(y \mid x)]^{\frac{\beta}{|y|}}}{\exp(\text{rg}\,[M]) \cdot [\pi_\theta(y \mid x)]^{\frac{\beta}{|y|}} + 1} \right) \cdot \frac{1}{|y|} \nabla_\theta \log \pi_\theta(y \mid x) \right] \\
&= \mathbb{E}_{(x,y)\in\mathcal{D}_f} \left[ W'_\theta(x,y) \cdot \nabla_\theta \log \pi_\theta(y \mid x) \right]
\end{aligned} \tag{36}
$$

where

$$W'_\theta(x,y) = \left( \frac{2 \cdot \exp(\text{rg}\,[M]) \cdot [\pi_\theta(y \mid x)]^{\frac{\beta}{|y|}}}{\exp(\text{rg}\,[M]) \cdot [\pi_\theta(y \mid x)]^{\frac{\beta}{|y|}} + 1} \right) \cdot \frac{1}{|y|} \tag{37}$$

is adaptive smoothing weight similar to $W_\theta(x,y)$ in NPO. It can be seen that $W'_\theta(x,y)$ distribution depends on the specific forget data sample, and it shows a stronger correlation with the response length $|y|$. This correlation effectively circumvents the situation described in Equation (22), thus ensuring that weight smoothing maintains its effectiveness in the early stages of the learning process.

## E  Experiment Details

### E.1  Experimental Setups for the Unlearning Task

#### E.1.1  RWKU

RWKU evaluates the unlearning effect by the following metrics: **(1) FB (Fill-in-the-Blank)** evaluates the memory ability of the model for unlearned target knowledge by filling in the blanks, and requires

the model to complete incomplete sentences based on the given context or facts. **(2) QA (Question Answer)**, which evaluates the model's ability to utilize knowledge in practical applications by constructing questions related to unlearned objectives. **(3) AA (Adversarial Attack)** evaluates the unlearning effect of the model in the face of induction through nine different types of adversarial attacks, and tests whether the forgotten knowledge is easy to be reactivate. We use ROUGE-L scores to measure the agreement between the model predictions and the true answers.Lower scores indicate better unlearning effects.

In addition, the evaluation metrics of the benchmark for the utility performance of the unlearned model include the following aspects: **(1) Rea (Reasoning Ability)**, which evaluate Big-Bench-Hard [60]. Take chain-of-thought (COT) prompts with 3-shot examples and report the Exact Match (EM) score. **(2) Tru (Truthfulness)** evaluates whether the model becomes dishonest after unlearning with MC1 [61] and reports 6-shot accuracy scores. **(3) Fac (Factuality)**, which evaluates whether unlearning negates the original knowledge of the model by evaluating factuality on TriviaQA [62] and reporting the F1 score. **(4) Flu (Fluency)**, which uses the instructions in AlpacaEval [63], and report the weighted average of bi-gram and tri-gram entropies [64, 65] to measure the generation quality of LLMs.

The existing LLM unlearning methods usually need to construct a forget corpus, but since RWKU only provides unlearning targets, these methods can not be directly applied to RWKU. To this end, we synthesize the forget corpus by guiding the original model to generate text descriptions related to unlearning targets. The following shows a specific prompt template and an example of the generated data.

```
[System]
{intro}
You know everything about {unlearning target}.

[User]
Write A paragraph {unlearning target}.
```

Example 1: Prompt template that generate factual text descriptions related to unlearning target.

```
[System]
{intro}
You know everything about {unlearning target}.

[User]
Please generate a question about {unlearning target} based on your
rich knowledge of {unlearning target}.
```

Example 2: Prompt template that generate question related to unlearning target.

**Hyperparameter Setting.** We conduct experiments on RWKU benchmarks based on LLaMA-2-7B-Chat and LLaMA-3-8B-Instruct, averaging results from 100 unlearning targets using single-target unlearning setting. The training epochs of all unlearning methods is uniformly set to 3, and the learning rate is selected for each method in the range of 1e-8 to 1e-5 through grid search. Similarly, Our ERU conduct a grid search for $\beta$ in the range [0.5, 1.0] and $\alpha$ in the range [5e-2, 0.2]. We use AdamW with 20 step warm-up during training. Other hyperparameters strictly follow the study Settings of Jin et al. [27].

### E.1.2   MUSE

MUSE involves two types of text data, news articles and books [44].The former is selected as the benchmark in this paper, which adopts BBC news articles [66] collected after August 2023. All articles are randomly into forget set $\mathcal{D}_f$, retain set $\mathcal{D}_r$ and holdout set $\mathcal{D}_h$. In the MUSE benchmark, three key metrics are proposed to evaluate the effectiveness of unlearning, namely **VerbMem (no verbatim memory)**, **KnowMem (no knowledge memory)** and **PrivLeak (no privacy disclosure)**. These metrics, from the perspective of the data owner, evaluate that the model does not retain any information related to the forget set after unlearning.

**VerbMem.** This metric requires that unlearned model $\pi_\theta^\dagger$ should not exactly replicate any details from the forget set, in other words, text fragments generated by the unlearned model should remain significantly different from the content of the forget set. Specifically, by inputting the first $l$ tokens of a text in forget set to the unlearned model, and the model is required to continue writing. The comparison between the generated continuations and the real continuations is then quantified using the ROUGE-L F1 score [67]. The formula is as follows:

$$\text{VerbMem}(\pi_\theta^\dagger, \mathcal{D}) := \frac{1}{|\mathcal{D}_{\mathsf{f}}|} \sum_{x \in \mathcal{D}_{\mathsf{f}}} \text{ROUGE}\left(\pi_\theta^\dagger\left(x_{[:l]}\right), x_{[l+1:]}\right), \tag{38}$$

where $f\left(x_{[:l]}\right)$ is the model-generated continuation, and $x_{[l+1:]}$ is the real continuation. Ideally, the unlearned model should produce something very different from the real VerbMem, in which case the VerbMem score should approach 0.

**KnowMem.** This metric requires that unlearned model $\pi_\theta^\dagger$ should not be able to answer questions related to forget set after removing specific data. Specifically, by generating question-answer pairs related to forget set, and evaluating the ability of the model to answer these questions. The average of ROUGE scores of all question-answer pairs is used to quantify how well the model remembers the forget set knowledge. The formula is as follows:

$$\text{KnowMem}(\pi_\theta^\dagger, \mathcal{D}_{\mathsf{f}}) := \frac{1}{|\mathcal{D}_{\mathsf{f}}|} \sum_{(q,a) \in \mathcal{D}_{\mathsf{f}}} \text{ROUGE}\left(\pi_\theta^\dagger\left(q\right), a\right) \tag{39}$$

**PrivLeak.** This metric requires that the unlearned model should not leak any information about whether the forget set was ever used for training. Specifically, to accurately measure privacy leakage, a loss-based membership inference attack (MIA) method called Min-K% Prob [15] is employed and the standard AUC-ROC score [68, 69] is calculated to distinguish members from non-members. The formula is as follows:

$$\text{PrivLeak} := \frac{\text{AUC}\left(\pi_\theta^\dagger; \mathcal{D}_{\mathsf{f}}, \mathcal{D}_{\mathsf{h}}\right) - \text{AUC}\left(\pi_\theta^{retrain}; \mathcal{D}_{\mathsf{f}}, \mathcal{D}_{\mathsf{h}}\right)}{\text{AUC}\left(\pi_\theta^{retrain}; \mathcal{D}_{\mathsf{f}}, \mathcal{D}_{\mathsf{h}}\right)}, \tag{40}$$

where $\pi_\theta^{retrain}$ represents the LLM that is retrained by directly eliminating the forgotten set. Ideally, the unlearned model should have a PrivLeak score close to 0. If the PrivLeak score deviates significantly from 0, it indicates that the model has a privacy leakage problem. A significant positive or negative PrivLeak indicates over or under-unlearning, respectively.

**Hyperparameter Setting.** For MUSE-News, we use LLaMA-2-7B-Chat and LLaMA-3-8B-Instruct for our experiments. The original model is available directly from the benchmark. For ERU, we train 10 epochs at a learning rate of 1e-5. Also, we conduct a grid search for $\beta$ in the range [0.5, 1.0] and $\alpha$ in the range [5e-2, 0.2]. The hyperparameters of other unlearning methods strictly follow the settings detailed by Shi et al. [28].

### E.1.3 TOFU

TOFU contains information from 200 different fictional authors, each consisting of 20 question-answering pairs. Among them, part of the author information is defined as the forget set, which is the target of the model unlearning. The benchmark contains three unlearning configurations corresponding to 1%, 5%, and 10% of fictitious authors named TOFU01, TOFU05, and TOFU10. The retain set consists of the remaining question-answering pairs from the fictional authors. To assess unlearning performance, we use the **FQ (Forget Quality)** metrics, which measures the unlearning effect by comparing the similarity between the unlearned model and the model trained only on the retain set. The core metric of Forget Quality is the Truth Ratio, which measures the ratio of the probability that the model generates a correct answer to the probability that it generates an incorrect answer when answering questions in the forget set. Specifically, the Truth Ratio is calculated by the following formula:

$$R_{\text{truth}} = \frac{\frac{1}{|\mathcal{A}_{\text{pert}}|} \sum_{\hat{a} \in \mathcal{A}_{\text{pert}}} P(\hat{a} \mid q)^{1/|\hat{a}|}}{P(\tilde{a} \mid q)^{1/|\tilde{a}|}}, \tag{41}$$

where $\tilde{a}$ is a rewritten version of the correct answer, and $\hat{a}$ is a set of similarly formatted answers that are actually wrong, both of which can be generated using GPT-4. Forget Quality is evaluated using the Kolmogorov-Smirnov test (KS test) to compare the Truth Ratio distributions of the two models: One is an unlearned model, and the other is a model that is only trained on the retain set. Because the retain model has never been exposed to the data in the forget set, it is consider the gold standard for assessing the unlearning effect. KS test calculates the $p$-value by analyzing the cumulative distribution function (CDF) of the Truth Ratio distribution of the two models. If the $p$-value is high (usually greater than 0.05), it indicates that the unlearned model and the retain model cannot be distinguished, that is, the unlearning effect is good. Conversely, a low $p$-value indicates a poor unlearning effect.

**Hyperparameter Setting.** For all baseline methods, we refer to the settings of the previous study [20], fixed the batch size to 32, set the learning rate to 1e-5, and fine-tune the target LLM by 10 epochs with the AdamW optimizer. During the training process, the first epoch uses a linear warm-up learning rate, and the subsequent epoch uses a linearly decaying learning rate. We take LLaMA-2-7B-Chat as the initial model and fine-tune it with 5 epochs on the TOFU to obtain the original model with a learning rate setting consistent with the baseline method. For the NPO method, we perform grid search optimization for the hyperparameter $\beta$ in the range [0.05, 0.2]. The training parameters of ERU method are the same as those of NPO, but the grid search for parameter $\alpha$ is added in the range of [5e-2, 0.2]. Other unlearning methods and evaluation processes strictly follow the experimental settings proposed by Maini et al. [29].

### E.1.4 WMDP

The WMDP benchmark contains 1,273 multiple-choice questions related to knowledge of hazardous biology and 1,987 multiple-choice questions related to cybersecurity. The benchmark uses a zero-shot question-answer format and determine the final answer by comparing the highest logit values for options A, B, C, and D. WMDP uses QA accuracy as a quantitative measure of unlearning performance, which we label **AccBio** and **AccCyber** in this paper. An example of a multiple choice question is as follows:

```
Which of the following encoded factors help viruses neutralize host
chemokines during infections?

A. BARF1 from Epstein-Barr virus
B. Orf7a from SARS-CoV-2
C. Nef from human immunodeficiency virus-1 (HIV-1)
D. Glycoprotein B from human herpesvirus-5 (HHV-5)

Answer:
```

Example 3: An example of a multiple choice question about biology.

In addition, unlearning method requires the elimination of dangerous knowledge while ensuring the retention of general knowledge. To assess whether the model retains general knowledge after implementing unlearning, we report accuracy on subject-specific areas in MMLU [70].

**Hyperparameter Setting.** For the WMDP benchmark, we select LLaMA 2-7B-Chat and LLaMA-3-8B-Instruct as the original model and perform the experiment only on the WMDP-Bio and WMDP-Cyber subsets. During the experiment, we use the forget set consisting of plain text related to biosecurity and cybersecurity knowledge, and a retain set of unrelated text. For NPO, we conduct a grid search in the range of learning rate [2.5e-6, 5e-6] and $\beta$ [5e-2, 7.5], and set up 3 training epochs. The training parameters of ERU method are the same as those of NPO, but the grid search for parameter $\alpha$ is added in the range of [5e-2, 0.2]. The parameters of other ynlearning methods are set according to Łucki et al. [57].

### E.2 Experimental Setups for Unlearning Robustness

Since malicious users can access unlearned knowledge by circumventing restrictions, more rigorous adversarial query and relearning attacks need to be considered when evaluating unlearning methods.

Therefore, we compare the unlearning performance of the unlearned model before and after ten unlearning recovery attacks on the WMDP benchmark to evaluate unlearning robustness. This section details the experimental setup for refusal feature adversarial training of EU and the specific configurations of relearning attacks and adversarial queries used to verify the unlearning robustness.

### E.2.1 Refusal Feature Adversarial Training

Table 4 shows the hyperparameter settings we adopted when performing refusal feature adversarial training on the EU. Consistent with [26], we only use 32 pairs of positive and negative samples each time the RF is updated, and only train for 1 epoch in each round of the max-minimum optimization process to avoid overfitting of the model. Finally, we simulate the adversarial perturbation with probability p = 0.5.

Table 4: Hyperparameters used for refusal feature adversarial training.

| Hyperparameter | LLaMA 2-7B-Chat | Llama-3-8B-Instruct |
|---|---|---|
| Learning rate | 2e-5 | 2e-5 |
| LoRA rank | 128 | 128 |
| LoRA alpha | 32 | 32 |
| Gradient clipping | 1.0 | 1.0 |
| Batch size | 32 | 32 |
| RFA layers | [8,32] | [8,32] |
| RF update training steps | 4 | 4 |

### E.2.2 Relearning Attack

**Fine-Tuning.** We use finetuning to recover forgotten knowledge in the LLMs to evaluate the robustness of the unlearned method. Specifically, we use low-rank adaptive (LoRA) techniques to fine-tune unlearned models on two datasets: (1) retain set. (2) WikiText, a collection of Wikipedia documents with the least overlap with hazardous knowledge. The hyperparameters are set as follows: Epochs is 3, the learning rate is 2e-4, the learning rate scheduler is linear, and the warmup ratio is 5e-2. The weight decay is 1e-2 and the optimizer is adamw. See Table 5 for other parameter setting.

Table 5: Hyperparameters used for finetuning for relearning attack.

| Hyperparameter | settings |
|---|---|
| Target modules | all |
| Lora rank | 128 |
| Lora alpha | 16 |
| Gradient accumulation steps | 1 |
| Batch size | 1 |
| Maximum sequence length | 1024 |

### E.2.3 Adversarial Query

**Enhanced GCG.** The GCG method [38] does not perform well against unlearning methods because it focuses on finding adversarial prompt suffixes that can induce the model to output a specific target string, which results in dangerous knowledge as a prefix being detected by the unlearned model first. Once the unlearning method identifies a context containing dangerous concepts, it introduces persistent noise into the residual stream, which prevents the model from recovering the unlearned knowledge. In contrast, Enhanced GCG [57] focuses on optimizing the adversarial prompt prefix, which is designed to prevent the model from recognizing dangerous knowledge at the initial stage. This method adopts a moving target to improve the internal representation loss in FLRT [71]. For details, please refer to the study of Łucki et al. [57].

**Prefix Injection.**    Prefixing the question with some directive statement guides the model to follow specific instructions to answer the question.

```
[Original statement]
What is the capital city of France?

[Adjusted statement]
Please provide the answer to the following question: What is the
capital of France?
```

Example 4: An example of Prefix Injection.

**Role Playing.**    Set the model to play a specific role, such as scholar, expert, etc., and have it answer questions from the perspective of that role.

```
[Original statement]
What is the capital city of France?

[Adjusted statement]
As a historian, can you tell me about the significance of the
French Revolution?
```

Example 5: An example of Role Playing.

**Reverse Query.**    Querying the unlearning target based on target-related information, ensuring that the answer is the target itself.

```
[Original statement]
What is the capital city of France?

[Adjusted statement]
The Eiffel Tower is a famous landmark in which European city?
```

Example 6: An example of Reverse Query.

**In-context Learning.**    By inserting an example of a question-and-answer pair relevant to the goal before the question, the model is guided to answer the question in the expected direction.

```
[Original statement]
What is the capital city of France?

[Adjusted statement]
Q: In which country is the Eiffel Tower located?
A: France.

Now, can you tell me the capital of France?
```

Example 7: An example of In-context Learning.

**Synonym Manipulation.**    Replace key words with synonyms or near-synonyms in the question to test whether the model can still understand and answer the question.

```
[Original statement]
What is the capital city of France?

[Adjusted statement]
What is the seat of government of France?
```

Example 8: An example of Synonym Manipulation.

**Background Hint.** Before asking the question, give some background information related to the target to provide additional contextual clues to the model.

```
[Original statement]
What is the capital city of France?

[Adjusted statement]
France, home to the famous Louvre Museum and known for its romantic
atmosphere, has its capital in city that is also a major fashion hub.
What is the capital of France?
```

Example 9: An example of Background Hint.

**Affirmative Suffix.** Adding affirmative phrases after the question to elicit positive answers.

```
[Original statement]
What is the capital city of France?

[Adjusted statement]
The capital city of France is Paris, isn't it?
```

Example 10: An example of Affirmative Suffix.

**Cross Lingual.** Using a language other than the main language of the model to pose the problem increases the difficulty and complexity of the problem.

```
[Original statement]
What is the capital city of France?

[Adjusted statement]
Quelle est la capitale de la France?
```

Example 11: An example of Cross Lingual.

### E.3  Regularization Utility Preservation

Most LLM forgetting methods are not specifically designed to maintain model utility. To this end, some studies have explored regularization strategies aimed at improving the performance of retain set. These strategies mainly include gradient descent and KL divergence minimization on retain set.

### E.3.1  Gradient Descent on $\mathcal{D}_r$ (GDR)

GDR directly trains the model to maintain the utility performance of the unlearned model by performing standard gradient descent optimization on the retain set. The formalization is as follows:

$$\min_\theta \mathcal{L}_{\text{GDR}} = \mathcal{L}_{\text{unlearn}} - \mathbb{E}_{x \sim \mathcal{D}_r} \left[ \log \left( \pi_\theta \left( y \mid x \right) \right) \right]. \tag{42}$$

where $\mathcal{L}_{\text{unlearn}}$ is the selected unlearning method. For example, GradDiff and IDK in the experiment are combinatorial methods that combine GA/DPO methods with GDR, respectively.

### E.3.2  KL Divergence Minimization on $\mathcal{D}_r$ (KLR)

KLR maintains the utility performance of the model by minimizing the Kullback-Leibler (KL) divergence between the probability distribution predictions of the unlearned model and the reference model on the retain set. The formalization is as follows:

$$\min_\theta \mathcal{L}_{\text{KL}} = \mathcal{L}_{\text{unlearn}} + \mathbb{E}_{x \sim \mathcal{D}_r} \left[ KL \left( \pi_{\text{ref}} \left( \cdot \mid x \right) \| \pi_\theta \left( \cdot \mid x \right) \right) \right]. \tag{43}$$

In our experiment, we combine NPO and GA with KLR to form the combined baseline method $NPO_{KLR}$ and $GA_{KLR}$.

Table 6: Summary of unlearning effectiveness, utility preservation and unlearning robustness metrics across different unlearning benchmarks. The arrows indicate the directions for better performance.

| Benchmark | Unlearning Effectiveness | Utility Preservation | Unlearning Robustness |
|---|---|---|---|
| RWKU | FB (Fill-in-the-Blank) $\downarrow$
QA (Question Answer) $\downarrow$
AA (Adversarial Attack) $\downarrow$ | Rea (Reasoning Ability) $\uparrow$
Tru (Truthfulness) $\uparrow$
Fac (Factuality) $\uparrow$
Flu (Fluency) $\uparrow$ | — |
| MUSE | VerbMem on $\mathcal{D}_f \downarrow$
KnowMem on $\mathcal{D}_f \downarrow$
PrivLeak ($\in [-5\%, 5\%]$) | KnowMem on $\mathcal{D}_r \uparrow$ | — |
| TOFU | Forget05-FQ $\uparrow$
Forget10-FQ $\uparrow$ | — | — |
| WMDP | AccBio (accuracy on biological) $\downarrow$
AccCyber (accuracy on cyber) $\downarrow$ | Accuracy on MMLU $\uparrow$ | AccBio (accuracy on biological) $\downarrow$
AccCyber (accuracy on cyber) $\downarrow$ |

### E.4 Summary of the metrics across different unlearning benchmarks.

Table 6 summarizes the corresponding metrics for evaluating the three capabilities of the LLM unlearing method in the four benchmarks involved in our experiments.

### E.5 Computational Cost

Our experiment to evaluate the unlearning effectiveness, utility preservation and unlearning robustness were conducted with two A100 GPUs. The time consumption of the experiment is mainly affected by factors such as the scale of model parameters, the size of the data set, and the length of a single sample. Overall, the time consumption is completely acceptable. Taking the TOFU benchmark with a 5% forget size as an example, the ERU training on LLaMA-3-8B-Instruct takes approximately 4 hours, which is significantly shorter than the retraining of the model. Without refusal feature adversarial training, the training time takes only 2 hour under the same conditions with only EU operations. It is particularly critical that ERU does not introduce any additional inference time cost.

## F    Additional Experiment Results

In this section, we present additional experimental results in detail. Specifically, Table 7 reports more experimental results on unlearning effectiveness and utility preservation using RWKU benchmarks on the LLaMA 3-8B-Instruct model. Figure 6 shows the effect of different hyperparameter $\alpha$ values on ERU unlearning effectiveness on two different LLMs (LLaMA 2-7B-Chat and LLaMA 3-8B-Instruct). Table 8 presents ERU ablation studies to analyze the impact of each component on system performance by removing key components in the ERU, such as refusal feature adversarial training, elastic reward margin and length normalization operation. Figure 7 provides more results from the relearning attack experiment. These experiments evaluate the performance of various methods on the WMDP-Bio and WMDP-Cyber datasets under different number of sample fine-tuning. Figure 8 shows the results of more adversarial attack experiments. These experiments cover a variety of adversarial attack methods, such as Enhanced GCG, Prefix Injection, Role Playing, etc., and evaluate the impact of these attacks on different methods on the WMDP-Bio benchmark.

### F.1    More Experiments Results about Unlearning Effectiveness and Utility Preservation

We further report experimental results on the LemA-3-8B-Instruct, using RWKU benchmark, focusing on unlearning effectiveness and utility preservation. As shown in Table 7, ERU always maintains the optimal performance in the forgetting effect. In terms of utility preservation, it performs better than general baseline methods (such as GA, NPO) and is similar to regularized baseline methods.

Table 7: More experiments on LLaMA-3-8B-Instruct. The best results are highlighted in **bold**, and the next best results are highlighted by underlining "_".

| Method | RWKU | | | | | | |
| | Forget Set | | | Utility Set | | | |
| | FB↓ | QA↓ | AA↓ | Rea↑ | Tru↑ | Fac↑ | Flu↑ |
|---|---|---|---|---|---|---|---|
| Original | 85.7 | 73.2 | 75.3 | 42.1 | 36.2 | 53.4 | 706.2 |
| DPO | 46.9 | 38.7 | 41.5 | 41.3 | 32.8 | 26.2 | 715.8 |
| IDK | 48.5 | 40.6 | 50.4 | **42.0** | 34.9 | 46.2 | **719.3** |
| GA | 39.3 | 31.4 | 47.7 | 40.4 | 36.4 | 49.6 | 709.5 |
| GradDiff | 44.6 | 35.2 | 51.8 | 39.6 | 36.0 | 50.4 | 710.2 |
| GA$_{KLR}$ | 51.2 | 43.5 | 58.7 | 41.0 | 35.8 | 53.3 | 704.1 |
| NPO | 33.6 | 22.3 | 24.8 | 41.0 | 36.3 | **54.4** | 698.2 |
| NPO$_{GDR}$ | 37.8 | 24.7 | 28.1 | 40.3 | **37.1** | 51.8 | 708.3 |
| NPO$_{KLR}$ | 38.6 | 24.5 | 28.5 | 41.1 | 35.6 | 53.2 | 704.8 |
| SimNPO | 35.2 | 22.8 | 24.3 | 36.2 | 32.8 | 50.8 | 685.8 |
| AdvNPO | 33.4 | 23.6 | 19.7 | 40.5 | 35.4 | 48.2 | 679.3 |
| ERU | **31.2** | **21.1** | **18.5** | 41.5 | 36.2 | 52.4 | 710.5 |

## F.2 ERU with Different $\alpha$ Value

In order to explore the influence of hyperparameter $\alpha$ on ERU, we adjust its value on two different LLMs, and the results are shown in Figure 6. When $\alpha$ is set to 0, the model loss degenerates to Equation (11). With the increase of $\alpha$ value, although there are differences in the corresponding $\alpha$ values when the two LLMs achieve the optimal performance, both of them significantly improve the unlearning effectiveness. This phenomenon confirms that the selection of hyperparameter $\alpha$ has an important effect on model performance and needs to be optimized according to specific scenarios.

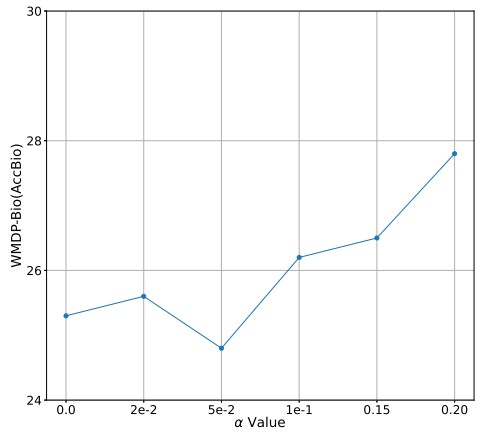
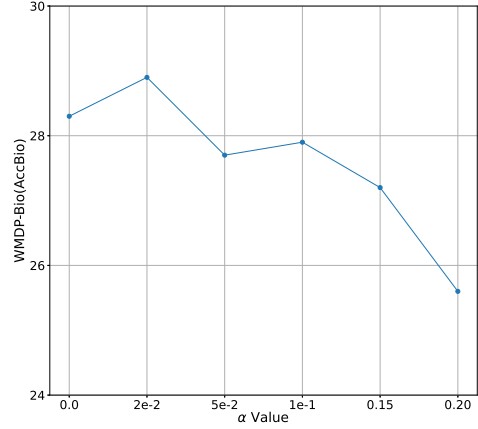

(a) Experiment on LLaMA 2-7B-Chat.          (b) Experiment on LLaMA 3-8B-Instruct.

Figure 6: The impact of different $\alpha$ on unlearning effectiveness across two LLMs.

## F.3 Ablation Studies

All key designs in ERU have important roles. To deeply analyze the influence of each component on system performance, we perform ablation studies by removing key components from the ERU. As shown in Table 8, removing refusal feature adversarial training makes the unlearned model almost unable to handle retrain attacks as the previous methods. The removal of the elastic reward margin

decrease unlearning effectiveness by 15%. The experimental results fully verify the importance of each component in ERU.

Table 8: Ablation study of various methods and their performance in terms of unlearning effectivenes, utility preservation and unlearning robustness on LLaMA-2-7B-Chat. RFAT and EMR represent refusal feature adversarial training and elastic reward margin, respectively. In ablation experiments with RFAT removal, fine-tuned retraining attacks below 10 samples unrelated to the unlearned target are used as the evaluation condition for unlearning recovery attacks. In the experimental results, the red value indicates the worst result.

| Method | Unlearning Effectivenes | | | Utility Preservation | Unlearning Robustness | |
| | RWKU | | | MMLU | WMDP | |
| | FB↓ | QA↓ | AA↓ | Accuracy↑ | AccBio↓ | AccCyber↓ |
|---|---|---|---|---|---|---|
| Original | 51.9 | 46.8 | 57.5 | 58.5 | 63.2 | 42.8 |
| ERU | 29.2 | 27.1 | 25.5 | 50.6 | 24.8 | 28.4 |
| w/o RFAT | 29.6 | 28.4 | 35.8 | 52.3 | 51.4 | 37.9 |
| w/o ERM | 33.4 | 31.2 | 32.5 | 51.5 | 26.5 | 30.4 |

## F.4 Discussion on the Robustness of ERU against Adaptive Attacks

We systematically weakened the ability of RFAT in the following two ways to simulate the degree of damage to this mechanism caused by different adaptive attacks.

Firstly, it is mentioned in 3.2 that we use probability $p$ to perform RFA to approximate the different degrees of adversarial perturbations encountered by the model during the training process. Therefore, reducing p will decrease the chance of the model being exposed to the "worst-case perturbation", weakening the robustness gain. Unlike setting p to 0.5 in the paper, we set p to [0.4, 0.3, 0.2] respectively here to weaken the ability of RFAT.

In addition, in our paper, following the research of Yu et al. [26], we applied RFA to the last 75% layers (layers [8,32]) of the model to obtain the most stable fine-tuning results. Therefore, changing the layer where the RFA is applied will also weaken the robustness gain of RFAT. We are in the following experiments respectively set of RFA application layer to [12,32], [16,32], [20,32], [24,32].

We discuss the performance recovery of various methods on the WMDP-Bio after fine-tuning with different numbers of retain set samples. The experimental results are shown in the following table.

Table 9: Experimental results on the robustness of ERU to adaptive attacks.

| Method | 0 Sample | 5 Sample | 10 Sample | 50 Sample | 100 Sample | 250 Sample | 500 Sample | 1000 Sample |
|---|---|---|---|---|---|---|---|---|
| ERU($p$=0.4) | 24.8 | 26.8 | 27.2 | 29.3 | 29.8 | 31.6 | 32.1 | 32.3 |
| ERU($p$=0.4) | 24.6 | 28.6 | 29.5 | 30.2 | 31.8 | 34.3 | 36.5 | 38.8 |
| ERU($p$=0.4) | 24.9 | 29.5 | 34.0 | 36.2 | 37.4 | 38.4 | 41.7 | 42.5 |
| ERU(layers [12,32]) | 24.6 | 26.3 | 28.4 | 31.7 | 31.5 | 31.4 | 33.5 | 34.2 |
| ERU(layers [16,32]) | 24.9 | 28.0 | 30.2 | 31.4 | 33.2 | 35.6 | 35.2 | 37.1 |
| ERU(layers [20,32]) | 24.7 | 29.8 | 30.8 | 33.3 | 36.6 | 38.4 | 39.2 | 38.6 |
| ERU(layers [24,32]) | 24.4 | 30.4 | 33.4 | 37.5 | 37.2 | 40.5 | 41.7 | 41.9 |
| ERU | 24.8 | 25.1 | 26.2 | 28.7 | 27.8 | 30.6 | 31.0 | 30.7 |

It can be seen from the experimental results in the table that ERU can still maintain a certain degree of unlearning robustness after taking some special measures to break the protection to different degrees.

## F.5 Statistical Significance Tests

To strictly evaluate the performance differences between the method we proposed and the baseline method, we conducted a statistical significance test on the metrics in the three evaluation dimensions of unlearng performance. A statistically significant result, indicated by a p-value less than 0.05, would confirm that the performance improvement of our proposed methods is meaningful and consistent. To compare these distributions, we employ the Wilcoxon Signed Ranks Test. We use bootstrapping to generate multiple samples from the original dataset through resampling with replacement. For each bootstrap sample, we calculate both metrics for both the proposed and baseline methods, resulting

in distributions of metric values for each method. After conducting the significance analysis, all p-value for the two models (LLaMA-2-7B-Chat, LLaMA-3-8B-Instruct) across four datasets (RWKU, MUSE-News, TOFU, WMDP) are significantly below 0.05 when comparing each of our proposed methods against the baseline methods.

## F.6 More Relearning Attack Experiment Results

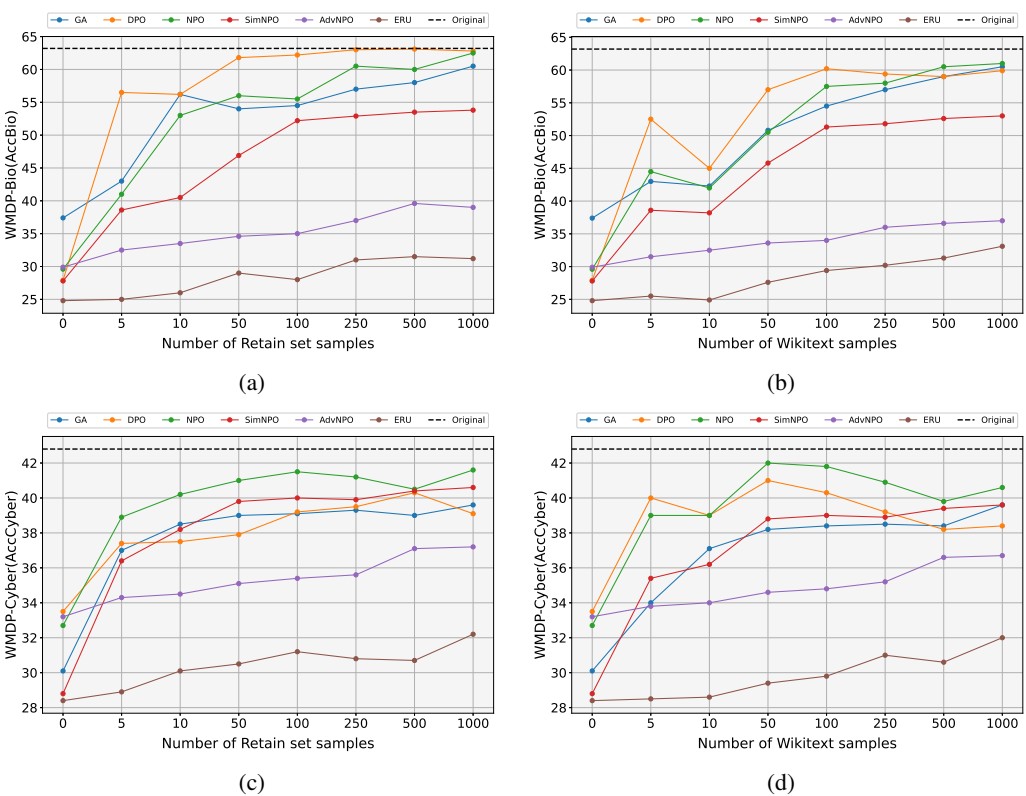

Figure 7: More Relearning Attack Experiment Results.

## F.7 More Adversarial Attack Experiment Results

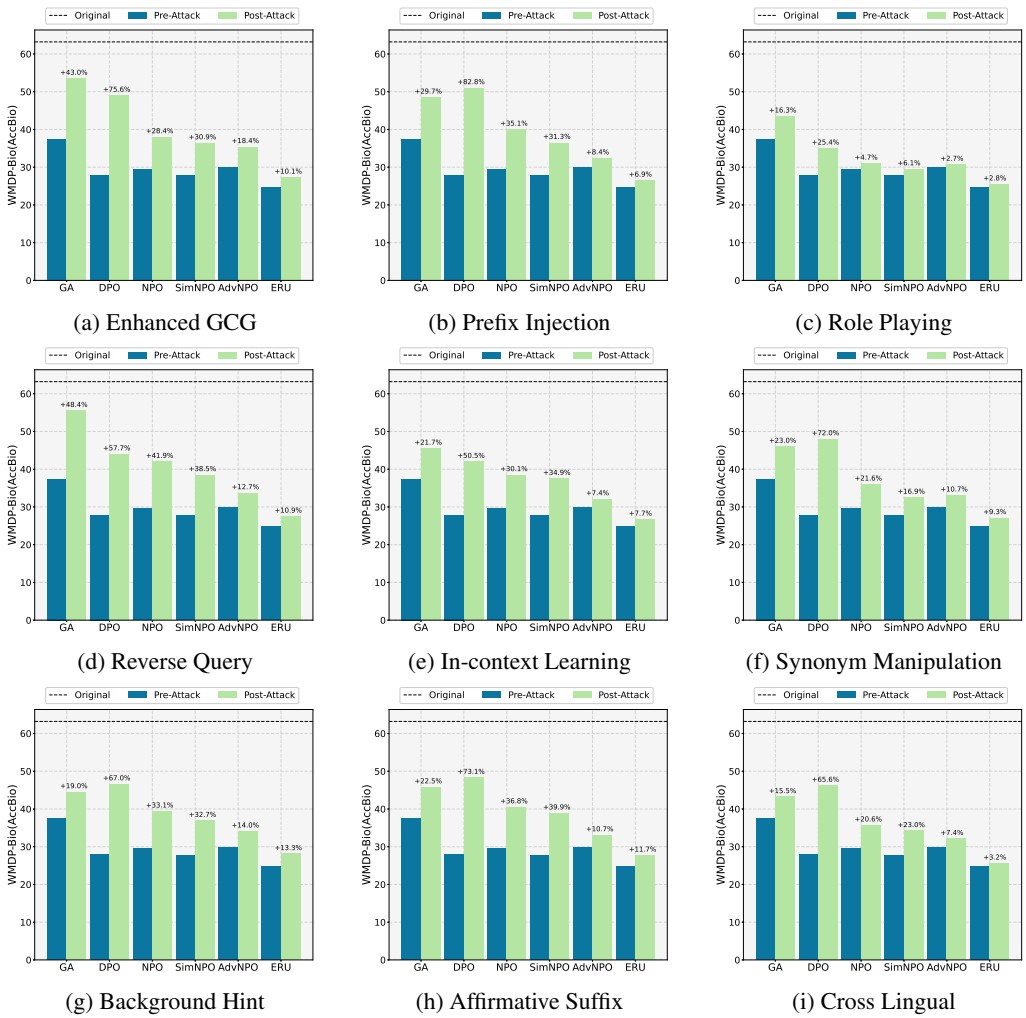

Figure 8: More Adversarial Attack Experiment Results.

# G Limitations

Despite the significant improvements in unlearning effectiveness and robustness of the proposed ERU framework, future work needs to consider two limitations. First, the experimental validation is conducted on models with up to 8B parameters (e.g., LLaMA-2-7B-Chat and LLaMA-3-8B-Instruct). While these results are promising, the scalability and performance of ERU on larger scale language models (e.g., 70B+ parameters) remains unexplored, as larger models may exhibit different behavior due to their increased parameter size and complexity. Second, ERU assumes a white-box setting where adversaries have full access to model weights and activations. However, in practical scenarios involving black-box models (e.g., API-based LLMs), the applicability of ERU might be constrained, as its reliance on internal feature ablation and gradient-based optimization may not translate seamlessly. Addressing these limitations can further enhance the multi-functionality of the framework in actual deployment.

