# OpenReview forum: "Elastic Robust Unlearning of Specific Knowledge in Large Language Models"
_NeurIPS.cc/2025/Conference — NeurIPS 2025 poster_

### Official Review · Reviewer_pWXE · 2025-06-26

**Clarity:** 3
**Significance:** 2
**Originality:** 2
**Rating:** 4
**Confidence:** 4

**Summary:**

This paper proposes Elastic Robust Unlearning (ERU), an optimization framework for robust and effective unlearning in Large Language Models. ERU introduces Elastic Reward Setting to balance reference-based and reference-free reward signals, providing flexible and adaptive optimization during unlearning. It utilizes Refusal Feature Ablation, which simulates worst-case perturbations during unlearning to defend against knowledge recovery preemptively. The paper formulates unlearning as a max-min optimization problem, where the inner loop simulates adversarial conditions, and the outer loop removes harmful knowledge.

**Questions:**

Could the authors compare the proposed method against other sophisticated adversarial training techniques under the same Elastic Reward settings? This would help isolate the value of the framework's adversarial component and demonstrate the rationale and whether its integration offers a meaningful improvement

Could the authors conduct an ablation study that evaluates the method using only the Elastic Reward component? This would clarify its individual contribution to overall performance and better support the design choices.

Could the authors perform statistical significance tests to demonstrate that the proposed method yields significant improvements in unlearning effectiveness while preserving model utility as claimed? This would strengthen the empirical claims and enhance the credibility of the reported gains.

**Ethical Concerns:**

["NO or VERY MINOR ethics concerns only"]

**Final Justification:**

The responses address my concerns related to ablation studies and statistical significance. Hence, I increase my scores.

**Quality:**

2

**Strengths And Weaknesses:**

Strengths:
1. ERU models the unlearning process as a robust optimization problem with Elastic Reward Setting to overcome the rigidity.

Weaknesses:
1. The proposed method incorporates a simplified adversarial training component, following a prior approach. However, it does not compare it to more advanced or state-of-the-art adversarial training techniques, instead of AdvNPO, to show the rationale for using Refusal Feature Ablation over others. Such comparisons are important for isolating the benefits of integrating adversarial mechanisms into the framework. Additionally, training time is typically not the primary concern in most machine unlearning scenarios.

2. The current ablation study lacks a variant that isolates the contribution of the Elastic Reward component. Evaluating the model's performance with only Elastic Reward (and without other components) would help quantify its individual impact and clarify the source of improvements.

3. The paper does not report statistical significance tests to support claims of performance improvement. Without such analysis, it is difficult to determine whether the observed gains are meaningful or within the margin of variability.

---

> ### Author Rebuttal · Authors · 2025-07-26
>
> Dear reviewer pWXE
>
> Thank you for your positive assessment of novelty and motivation of our work. We hope to address your concerns in our reply below.
>
> ###  **Weaknesses and Questions:**
> > **Weakness (1) and Question  (1)** :Could the authors compare the proposed method against other sophisticated adversarial training techniques under the same Elastic Reward settings? [...]
>
> Thank you for bringing it to our attention! We supplement the comparison of the proposed method against Input-sapce Adversarial Training (IAT) **[1]**, Continuous Adversarial Training (CAT) **[2]** and Latent-space Adversarial Training (LAT) **[3]** under the same Elastic Reward settings. The experimental results are shown in three dimensions as follows:
>
> `Unlearning Effectivenes`
> |Method|RWKU|RWKU|RWKU|MUSE-News|MUSE-News|MUSE-News|TOFU|TOFU|WMDP|WMDP|
> |  :----  | :----: | :----: | :----: | :----: | :----: | :----: | :----: | :----: | :----: | :----: |
> ||FB↓| QA↓ |AA↓|VerbMem↓ |KnowMem↓ |PrivLeak    (∈ [−5%, 5%])|Forget05-FQ↑| Forget10-FQ↑|AccBio↓ | AccCyber↓|
> |EU-IAT|34.7|32.2|30.5|15.7|14.8|`27.8`|0.58|0.32|29.5|31.5|
> |EU-CAT|33.5|`29.4`|28.7|13.1|17.8|34.6|0.62|0.41|`29.2`|31.2|
> |EU-LAT|`32.8`|29.6|`25.`8|`12.5`|`11.8`|31.5|`0.71`|`0.44`|`27.3`|`29.5`|
> |ERU|**29.2**| **27.1**| **25.5**| **10.4** |**9.2** |**12.3**| **0.73**| **0.48**|**24.8**| **28.4**|
>
> `Utility Preservation`
> |method|RWKU|RWKU|RWKU|RWKU|MUSE-News|MMLU|
> |  :----  | :----: | :----: | :----: | :----: | :----: | :----: |
> ||Rea↑ |Tru↑ |Fac↑ |Flu↑| KnowMem↑|Accuracy↑|
> |EU-IAT|24.3 |28.9| 37.5 |685.4|36.9|40.8|
> |EU-CAT|`25.2` |`29.4`| `39.2` |`697.3`|40.8|42.5|
> |EU-LAT|24.8 |28.7| 38.8 |688.5|`41.6`|`45.7`|
> |ERU|**26.2** |**30.5**| **40.5**|**708.8** |**43.2**|**50.6**|
>
> `Unlearning Robustness` (Performance recovery of various methods on the WMDP-Bio after fine-tuning with different numbers of retain set samples.)
> |Method|0 samples|5 samples|10 samples|50 samples|100 samples|250 samples|500 samples|1000 samples|
> |  :----  | :----: | :----: |  :----: |  :----: |  :----: |  :----: |  :----: |  :----: |
> |EU-IAT|29.5 | 33.5| 34.7|35.4|36.2|37.9|38.6|39.3|
> |EU-CAT|29.2 | 33.8| 34.6|34.9|35.0|35.8|37.6|38.4|
> |EU-LAT|`27.3` | `29.5`| `30.6`|`32.8`|`33.6`|`34.2`|`34.5`|`34.7`|
> |ERU|**24.8** | **25.1**| **26.2**|**28.7**|**27.8**|**30.6**|**31.0**|**30.7**|
>
> Experimental results show that ERU combined with RFAT significantly outperforms the methods combined with other adversarial training means. The best method is **bolded**, and the second-best method in `highlight`.
>
> ******
>
> > **Weakness (2) and Question  (2)** : Could the authors conduct an ablation study that evaluates the method using only the Elastic Reward component? [...]
>
> Thank you for your valuable suggestions. In fact, we have already conducted relevant discussions in **Appendix F.3** of the paper. In this section, we adopted the approach of removing the key components of EUR to deeply analyze the specific impact of different components on performance. The situation you mentioned, "only using the Elastic Reward component to evaluate the method", actually corresponds to the state after removing the refusal feature adversarial training (RFAT) from the ERU. The specific results can be referred to in the following table (consistent with **Table 8 in Appendix F.3** of the paper) :
>
> |Method|Unlearning Effectivenes|Unlearning Effectivenes|Unlearning Effectivenes|Utility Preservation|Unlearning Robustness|Unlearning Robustness|
> |  :----  | :----: | :----: | :----: | :----: | :----: | :----: |
> ||RWKU|RWKU|RWKU|MMLU|WMDP|WMDP|
> ||FB↓|QA↓|AA↓|Accuracy↑|AccBio↓|AccCyber↓|
> |Original|51.9|46.8|57.5|58.5|63.2|42.8|
> |ERU|**29.2** |**27.1**|**25.5**|50.6|**24.8**|**28.4**|
> |w/o RFAT (only EU)|29.6|28.4|35.8|**52.3**|51.4|37.9|
>
> We realize that this part of the content might not have been emphasized in the main text, which has raised your question. To make the presentation of viewpoints more clear, we plan to directly incorporate this part of the analysis into the main text rather than the appendix to ensure that readers can understand them more clearly.
>
> ******
>
> > **Weakness (3) and Question  (3)** : Could the authors perform statistical significance tests to demonstrate that the proposed method yields significant improvements in unlearning effectiveness while preserving model utility as claimed? [...]
>
> We sincerely appreciate the reviewer's valuable suggestion. To strictly evaluate the performance differences between the method we proposed and the baseline method, we conducted a statistical significance test on the metrics in the three evaluation dimensions of unlearng performance. A statistically significant result, indicated by a p-value less than 0.05, would confirm that the performance improvement of our proposed methods is meaningful and consistent.  To compare these distributions, we employ the **Wilcoxon Signed Ranks Test**. We use bootstrapping to generate multiple samples from the original dataset through resampling with replacement.  For each bootstrap sample, we calculate both metrics for both the proposed and baseline methods, resulting in distributions of metric values for each method. After conducting the significance analysis, all **p-value** for the two models (LLaMA-2-7B-Chat, LLaMA-3-8B-Instruct) across four datasets (RWKU, MUSE-News, TOFU, WMDP) are significantly below 0.05 when comparing each of our proposed methods against the baseline methods. The results of the statistical significance tests are as follows:
>
> `Unlearning Effectivenes`
> |Metrics|RWKU|RWKU|RWKU|MUSE-News|MUSE-News|MUSE-News|TOFU|TOFU|WMDP|WMDP|
> |  :----  | :----: | :----: | :----: | :----: | :----: | :----: | :----: | :----: | :----: | :----: |
> ||FB| QA |AA|VerbMem |KnowMem |PrivLeak|Forget05-FQ| Forget10-FQ|AccBio | AccCyber|
> |p-value (LLaMA-2-7B-Chat) ​|2.8e-3 |1.4e-3|3.8e-3|4.2e-4|9.6e-5|7.3e-3|1.9e-2|3.1e-2|2.1e-4|1.7e-3|
> |p-value (LLaMA-3-8B-Instruct) ​|3.4e-3 |1.6e-2|5.8e-3|2.4e-3|7.2e-4|2.4e-3|2.5e-2|1.3e-2|6.5e-4|5.8e-3|
>
> `Utility Preservation`
> |Metrics|RWKU|RWKU|RWKU|RWKU|MUSE-News|MMLU|
> |  :----  | :----: | :----: | :----: | :----: | :----: | :----: |
> ||Rea |Tru |Fac |Flu| KnowMem|Accuracy|
> |p-value (LLaMA-2-7B-Chat) ​​|2.7e-3 |8.9e-4| 1.8e-3| 3.1e-3 |1.2e-2 |1.3e-3|
> |p-value (LLaMA-3-8B-Instruct) ​​|1.8e-3 |3.6e-3| 1.9e-3| 3.5e-4|2.2e-3 |5.3e-4|
>
> `Unlearning Robustness`
> |Metrics|RWKU|RWKU|RWKU|WMDP|WMDP|
> |  :----  | :----: | :----: | :----: | :----: | :----: |
> ||FB|QA|AA|AccBio|AccCyber|
> |p-value​ (LLaMA-2-7B-Chat) ​|1.7e-2 |6.2e-3| 1.9e-4| 2.3e-3 |2.2e-3 |1.1e-2|
> |p-value (LLaMA-3-8B-Instruct) |1.6e-3 |9.2e-4| 8.5e-4| 1.3e-2 |7.2e-3 |3.5e-3|
>
>
> ### **Concluding remarks.**
>
> We would be grateful if you could let us know whether our explanations have addressed your concerns. Please let us know if you have any other questions or concerns.
>
> ### **References.**
> [1] Zou, et al. (2023) Universal and Transferable Adversarial Attacks on Aligned Language Models.
>
> [2] Xhonneux, et al. (2024) Efficient Adversarial Training in LLMs with Continuous Attacks.
>
> [3] Sheshadri, et al. (2024) Latent Adversarial Training Improves Robustness to Persistent Harmful Behaviors in LLMs.

---

> > ### Comment · Reviewer_pWXE · 2025-08-05
> >
> > The responses address my concerns. Hence, I increase my scores.

---

> > > ### Author Response · Authors · 2025-08-05
> > >
> > > Dear reviewer pWXE,
> > >
> > > Thanks for taking the time to review our work, we have carefuly considered your comments and made every efort to respond to your concerns.
> > >
> > > If you have any further questions or require additional clarification, please kindly let us know.
> > >
> > > Best regards.

---

### Official Review · Reviewer_uCL3 · 2025-07-02

**Clarity:** 3
**Significance:** 3
**Originality:** 3
**Rating:** 5
**Confidence:** 4

**Summary:**

This paper proposes Elastic Robust Unlearning (ERU), a novel framework for improving the effectiveness and robustness of large language model (LLM) unlearning. Addressing limitations in existing preference optimization (PO)-based methods, ERU introduces two key innovations: an elastic reward mechanism that enhances unlearning flexibility, and refusal feature ablation, which induces targeted failure modes to boost robustness against unlearned knowledge relearning. Experimental results demonstrate that ERU achieves superior unlearning performance while preserving model utility, outperforming prior methods in a number of benchmark datasets and tasks.

**Questions:**

Please see the above weaknesses.

**Ethical Concerns:**

["NO or VERY MINOR ethics concerns only"]

**Final Justification:**

After careful reading of the authors' response, my concerns have been addressed. The authors should incorporate the additional experiments and analysis into the future revision. I am glad to raise my rating to Accept.

**Limitations:**

yes

**Quality:**

3

**Strengths And Weaknesses:**

## Strengths
1. This work provides a detailed analysis of the shortcomings of existing LLM unlearning approaches.

2. The proposed methods smartly balance the weights of reference-based and reference-free policy optimization during unlearning.

3. Adopting the refusal feature ablation can greatly reduce the computation cost compared to using regular adversarial training to prevent unlearned knowledge relearning.

4. The paper is well-organized, and it is easy for the readers to capture the main ideas.

## Weaknesses

1. The refusal feature ablation (RFA) is directly adopted to enhance the robustness of elastic policy optimization. Dedicated design is required for the scenarios of LLM unlearning. For example, the D_harmful and D_harmless can be constructed by the forget or retained datasets. In real-world scenarios, unlearning may be employed to delete outdated information and users' private information.

2. More details of balancing the inner and outer optimization should be provided. For max-min bi-level optimization, the optimization pace impacts the final performance a lot.

3. There is no ablation study of detaching the refusal feature ablation from ERU. The balancing effect between reference-based and reference-free optimization offered by the elastic reward margin needs validation.

4. The robustness of defending against adaptive attacks should be discussed and validated. That is to say, if the adversaries have the prior knowledge that the target LLM has been protected by RFA and they adopt some dedicated measures to break up the protection before conducting unlearned knowledge relearning, how will ERU perform, how long, and how strong will the unlearning effect last?

---

> ### Author Rebuttal · Authors · 2025-07-28
>
> Dear reviewer uCL3
>
> Thank you for your constructive comments and they are valuable for improving our paper. In the following, we will address your concerns one by one.
>
> ###  **Weaknesses and Questions:**
>
> > **Weakness (1)**: The refusal feature ablation (RFA) is directly adopted to enhance the robustness of elastic policy optimization. Dedicated design is required for the scenarios of LLM unlearning. [...]
>
> Thank you for your valuable suggestions. In our current design, our D_harmful and D_harmless are respectively sampled from AdvBench and Alpaca, which is based on the existing practice in the study of refusal features [1]. This choice stems from the primary goal of our RFAT, which is to simulate the worst-case adversarial attacks designed to bypass the model's security mechanisms and trigger harmful outputs.
>
> We think your suggestion is very valuable. Constructing D_ harmful set directly from the forgotten set (D_f) and D_ harmless set from the retained set (D_r) is a very persuasive and conceptually elegant idea, which is particularly suitable for the unlearning task. This perfectly aligns with the core objective of this paper, that is, we hope that the model "refuses" to output or utilize the knowledge in D_f while retaining the knowledge in D_r and responding normally to it. We verify this suggestion by supplementing the following experiments:
>
> `Unlearning Effectiveness`
> |Method|RWKU|RWKU|RWKU|MUSE-News|MUSE-News|MUSE-News|TOFU|TOFU|WMDP|WMDP|WMDP|
> |  :----  | :----: | :----: | :----: | :----: | :----: | :----: | :----: | :----: | :----: | :----: | :----: |
> ||FB↓| QA↓ |AA↓|VerbMem↓ |KnowMem↓ |PrivLeak |Forget05-FQ↑| Forget10-FQ↑|AccBio↓ | AccCyber↓|  AccChem↓|
> |ERU|`29.2`|`27.1`| `25.5`| `10.4` |`9.2` |`12.3`| `0.73`| `0.48` |`24.8`| `28.4`|`27.2`|
> |ERU (Sample from D_f and D_r)|**28.4**|**26.7**| **25.3**| **9.8** |**9.1** |**11.5**| **0.85**| **0.49** |**24.4**| **27.5**|**26.3**|
>
> `Utility Preservation`
> |Method|RWKU|RWKU|RWKU|RWKU|MUSE-News|MMLU|TOFU-Forget05|TOFU-Forget05|TOFU-Forget10|TOFU-Forget10|
> |  :----  | :----: | :----: | :----: | :----: | :----: | :----: | :----: | :----: | :----: | :----: |
> ||Rea↑ |Tru↑ |Fac↑ |Flu↑| KnowMem↑|Accuracy↑|Probability↑|ROUGE↑|Probability↑|ROUGE↑|
> |ERU|`26.2` |`30.5`| **40.5**| `708.8` |`43.2` |`50.6`|`0.59`|**0.56**|`0.74`|`0.53`|
> |ERU(Sample from D_f and D_r)|**26.4** |**31.0**| `40.3`| **711.5** |**46.1** |**51.8**|**0.63**|**0.56**|**0.75**|**0.55**|
>
> `Unlearning Robustness`
> |Method|0 samples|5 samples|10 samples|50 samples|100 samples|250 samples|500 samples|1000 samples|
> |  :----  | :----: | :----: |  :----: |  :----: |  :----: |  :----: |  :----: |  :----: |
> |ERU|`24.8` | `25.1`| `26.2`|`28.7`|**27.8**|`30.6`|`31.0`|`30.7`|
> |ERU(Sample from D_f and D_r)|**24.4**|**24.7**|**25.8**|**27.9**|`28.2`|**29.7**|**29.6**|**30.1**|
>
> Experimental results show that, for different benchmarks, constructing D_harmful and D_harmless by sampling from its forget set and retain set respectively can better improve the ability of ERU, including Unlearning Effectiveness, Utility Preservation, and Unlearning Robustness.The best method is **bolded**, and the second-best method in `highlight`.
>
> > **Weakness (2)**: More details of balancing the inner and outer optimization should be provided. For max-min bi-level optimization, the optimization pace impacts the final performance a lot.
>
> Thank you for pointing this out! To provide more details on balancing the inner and outer optimization, we supplement the following experiments here. Specifically, we explore the influence of the number of inner optimization steps. We follow the same experimental configurations as in the paper, but instead vary the number of inner optimization steps. The experimental results are shown in the following table:
>
> |Method|Unlearning Effectiveness|Unlearning Effectiveness|Unlearning Effectiveness|Utility Preservation|
> |  :----  | :----: | :----: | :----: | :----: |
> ||RWKU|RWKU|RWKU|MMLU|
> ||FB↓|QA↓|AA↓|Accuracy↑|
> |ERU(Step=1)|29.6 |28.3|30.4|45.3|
> |ERU(Step=2)|29.4 |27.8|28.3|45.6|
> |ERU(Step=3)|29.1 |27.5|26.6|45.9|
> |ERU(Step=4)|28.9 |27.0|26.2|46.8|
> |ERU(Step=5)|29.1 |27.1|25.3|48.5|
> |ERU(Step=6)|29.2 |27.1|25.5|50.6|
> |ERU(Step=7)|30.5 |28.5|25.8|45.8|
> |ERU(Step=8)|30.2 |28.2|26.9|47.4|
>
> From the table, we can see that as the optimization steps increases, the Unlearing Effectiveness initially declines (this is what we hope to see), then rises. For Utility Preservation, this trend is rising first, and then decline. In addition, for the Unlearing Robustness, although the more sufficient inner loop will simulate the stronger adversaries and make the model obtain stronger robustness gain, it will greatly increase the burden of the outer loop and make the training time longer. We thereby conclude that both insufficient and excessive optimization steps are detrimental to unlearning performance. To better balance the performance of the ERU, we have set the inner optimization steps to 6 in the experiments of the paper.
>
>
> > **Weakness (3)**: There is no ablation study of detaching the refusal feature ablation from ERU. The balancing effect [...]
>
> We get your concern. In fact, we have already conducted relevant discussions in **Appendix F.3** of the paper. In Appendix F.3, we adopted the approach of removing the key components of ERU to deeply analyze the specific impact of different components on performance, including the ablation study of detaching the refusal feature ablation from ERU. The specific results can be referred to in the following table (consistent with Table 8 in Appendix F.3) :
>
> |Method|Unlearning Effectiveness|Unlearning Effectiveness|Unlearning Effectiveness|Utility Preservation|Unlearning Robustness|Unlearning Robustness|
> |  :----  | :----: | :----: | :----: | :----: | :----: | :----: |
> ||RWKU|RWKU|RWKU|MMLU|WMDP|WMDP|
> ||FB↓|QA↓|AA↓|Accuracy↑|AccBio↓|AccCyber↓|
> |Original|51.9|46.8|57.5|58.5|63.2|42.8|
> |ERU|**29.2** |**27.1**|**25.5**|50.6|**24.8**|**28.4**|
> |w/o RFAT|29.6|28.4|35.8|**52.3**|51.4|37.9|
>
> > **Weakness (4)**: The robustness of defending against adaptive attacks should be discussed and validated. [...]
>
> Thank you for putting forward this very valuable suggestion. To address your concerns, we add the following discussion on the robustness against adaptive attacks. Just as you said, "adopt some dedicated measures to break up the protection", we systematically weakened the ability of RFAT in the following two ways to simulate the degree of damage to this mechanism caused by different adaptive attacks.
>
> Firstly, the paper mentions that we use probability p to perform RFA to approximate the different degrees of adversarial perturbations encountered by the model during the training process. Therefore, reducing p will decrease the chance of the model being exposed to the "worst-case perturbation", weakening the robustness gain. Different from setting p to 0.5 in the paper, we set p to [0.4, 0.3, 0.2] respectively here to weaken the ability of RFAT.
>
> In addition, in our paper, following the research of Yu et al., we applied RFA to the last 75% layers  (layers [8,32]) of the model to obtain the most stable fine-tuning results. Therefore, changing the layer where RFA is applied will also weaken the robustness gain of RFAT. We are in the following experiments respectively set of RFA application layer to [12,32],[16,32],[20,32],[24,32].
>
> Consistent with the paper, We discuss the performance recovery of various methods on the WMDP-Bio after fine-tuning with different numbers of retain set samples.  The experimental results are shown in the following table:
>
> |Method|0 samples|5 samples|10 samples|50 samples|100 samples|250 samples|500 samples|1000 samples|
> |  :----  | :----: | :----: |  :----: |  :----: |  :----: |  :----: |  :----: |  :----: |
> |ERU(p=0.4)|24.8 | 26.8| 27.2|29.3|29.8|31.6|32.1|32.3|
> |ERU(p=0.3)|24.6 | 28.6| 29.5|30.2|31.8|34.3|36.5|38.8|
> |ERU(p=0.2)|24.9 | 29.5| 34.0|36.2|37.4|38.4|41.7|42.5|
> |ERU(RFA layers [12,32])|24.6 | 26.3| 28.4|31.7|31.5|31.4|33.5|34.2|
> |ERU(RFA layers [16,32])|24.9 | 28.0| 30.2|31.4|33.2|35.6|35.2|37.1|
> |ERU(RFA layers [20,32])|24.7 | 29.8| 30.8|33.3|36.6|38.4|39.2|38.6|
> |ERU(RFA layers [24,32])|24.4 | 30.4| 33.4|37.5|37.2|40.5|41.7|41.9|
> |ERU|24.8 | 25.1| 26.2|28.7|27.8|30.6|31.0|30.7|
>
> It can be seen from the experimental results in the table that ERU can still maintain a certain degree of unlearning robustness after taking some special measures to break the protection to different degrees.
>
>
> ### **Concluding remarks.**
>
> We would be grateful if you could let us know whether our explanations have addressed your concerns. Please let us know if you have any other questions or concerns.
>
> ### **References.**
>
> [1] Arditi, et al. (2024) Refusal in Language Models Is Mediated by a Single Direction.

---

> > ### Comment · Reviewer_uCL3 · 2025-08-06
> >
> > After careful reading of the authors' response, my concerns have been addressed. The authors should incorporate the additional experiments and analysis into the future revision. I am glad to raise my rating to Accept.

---

### Official Review · Reviewer_NnGx · 2025-07-03

**Clarity:** 3
**Significance:** 3
**Originality:** 2
**Rating:** 5
**Confidence:** 3

**Summary:**

This paper introduces Elastic  Robust Unlearning (ERU), a novel framework designed to remkove specific knowledge from LLM more effectively and more robustly than the PO-based unlerarning algorithms like DPO, NPO, or the variants of NPO. They have two key designs: 1. the elastic reward setting, by using a reference model value combined between the uniforms distribution and the original model's output. This is in contrast with the rigid reward setting in prior works that either uses a uniform distribution as the reference value or use the original model's prediction probability. 2. They apply a refusal feature ablation (RFA)-based adversarial training procedure during the training process to simulate adversarial removal of the refusal featurein the model's hidden activation and formulate the robust unlearning as a max-min optimization problem. The experiments on multiple unlearning benchmarks, such as TOFU, WMDP, MUSE, show that ERU outperforms PO-based algorithms by a good margin.

**Questions:**

See the Strength and Weakness Section.

**Ethical Concerns:**

["NO or VERY MINOR ethics concerns only"]

**Final Justification:**

Thanks the authors for their responses in detail. I read your response and I really appreciate it, especially for clarifying your contribution in more detail and providing more ablation studies to show the strength of the method, so I raise my scores. I also read other reviewers' responses and find them helpful.

**Limitations:**

See the Strength and Weakness Section.

**Paper Formatting Concerns:**

/

**Quality:**

2

**Strengths And Weaknesses:**

Strengths:

1. The paper evaluates ERU across multiple axes—effectiveness, utility preservation, and robustness. Experimental results are favorable across multiple benchmarks compared to the most widely used methods, such as GA and NPO, or their variants. This is convincing and shows the effectiveness of the designed algorithm.

2. The combination of the RFA-based adversarial training is novel and improved the robustness of unlearning algorithms. Incorporating RFA into the unlearning loop is a clever way to approximate worst-case adversarial perturbations without costly inner-loop PGD. By randomly ablating learned refusal features in the residual streams with probability p, ERU achieves robustness comparable to latent-space adversarial training but at a lower computational cost.

Weakenesses and Questions:

1. The novelty of this elastic reward setting: Given that ERU’s elastic reward formulation collapses to the rigid reference (only using uniform distribution or only using reference models) when alpha = 0 or alpha = 1, why should elastic and rigid reward methods be considered fundamentally distinct classes rather than simply two points on the same spectrum? In particular, since every rigid scheme can be recovered by fixing
alpha at an extreme, what principled argument or theoretical insight supports treating ERU as a qualitatively new paradigm?

2. You did some ablation studies about the effects of two main components of ERU: the elastic reward design and the RFA-based adversarial training in the appendix, but this ablation is only on the RWKU dataset for comparing the effectiveness. I wonder how they perform on the other three datasets.

3. Figure 6 (PS: It does not have a caption) shows the effect of alpha and shows that setting alpha away from zero will improve the performance. It seems that the optimal alpha highly depends on the model (what about across datasets? What are the optimal alpha?) Is there anyway to explain this? Do you have any more intuition about what the hyperparameter alpha is and what it controls? And the most importantly, in what case would you expect a larger optimal alpha?

---

> ### Author Rebuttal · Authors · 2025-07-29
>
> Dear reviewer NnGx
>
> Thank you for your positive evaluation of  the novelty of our work and the comprehensiveness of our experiments. We are pleased to address your concerns.
>
> ###  **Weaknesses and Questions:**
>
> > **Weakness and Question (1)**: The novelty of this elastic reward setting [...] why should elastic and rigid reward methods be considered fundamentally distinct classes rather than simply two points on the same spectrum? [...]  what principled argument or theoretical insight supports treating ERU as a qualitatively new paradigm?
>
> Thank you for your profound insights into the elastic reward settings in the ERU. You have correctly noticed that when α = 0, the elastic reward setting of ERU has been transformed into reference-free reward. However, what we need to clarify is that when α = 1, the elastic reward setting of ERU does not completely collapse into reference-based reward. At this time, the influence of uniform distribution $U (y | x)$ still exists (Equation 13 of the paper).
>
> Therefore, what we want to emphasize is that the elastic reward setting of ERU is not merely a simple hyperparameter interpolation. Instead, starting from the limitations of the rigid reward setting (Appendix B), and addressing these limitations by dynamically balancing the complementary advantages of reference-based reward and reference-free reward (Appendix D.2). The elastic reward setting is precisely the new category defined to distinguish this rigid setting. ERU converts rigid reward setting into a continuous and adjustable spectrum, in which α controls the degree of influence between reference and non-reference components, a paradigm that was lacking in previous work.
>
> *****
>
> > **Weakness and Question (2)**: You did some ablation studies about the effects of two main components of ERU [...] I wonder how they perform on the other three datasets.
>
> Thank you for your insightful observation regarding the scope of our ablation studies.   We appreciate the opportunity to address this limitation and provide comprehensive cross-dataset ablation results below:
>
> `Ablation results of Unlearning Effectiveness`
> |Method|RWKU|RWKU|RWKU|MUSE-News|MUSE-News|MUSE-News|TOFU|TOFU|WMDP|WMDP|WMDP|
> |:--|:--:|:--:|:--:|:--:|:--:|:--:|:--:|:--:|:--:|:--:|:--:|
> ||FB↓| QA↓ |AA↓|VerbMem↓ |KnowMem↓ |PrivLeak |Forget05-FQ↑| Forget10-FQ↑|AccBio↓ | AccCyber↓|  AccChem↓|
> |Original|51.9|46.8| 57.5| 58.3 |63.7 |-99.8| 3.2e-16| 2.1e-19 |63.2| 42.8|52.4|
> |ERU|**29.2**|**27.1**| **25.5**| `10.4` |**9.2** |**12.3**| **0.73**| `0.48` |**24.8**| **28.4**|**27.2**|
> |w/o RFAT|`29.6`|`28.4`| `35.8`| **10.2** |`9.3` |`21.7`| `0.71`| **0.49** |`26.5`| `28.3`|`29.8`|
> |w/o ERM |33.4|31.2| 32.5| 12.3 |11.1 |18.4| 0.64| 0.44 |28.2| 29.4|30.5|
>
> `Ablation results of Utility Preservation`
>
> |Method|RWKU|RWKU|RWKU|RWKU|MUSE-News|MMLU|TOFU-Forget05|TOFU-Forget05|TOFU-Forget10|TOFU-Forget10|
> |:--|:--:|:--:|:--:|:--:|:--:|:--:|:--:|:--:|:--:|:--:|
> ||Rea↑ |Tru↑ |Fac↑ |Flu↑| KnowMem↑|Accuracy↑|Probability↑|ROUGE↑|Probability↑|ROUGE↑|
> |Original|26.9 |30.4| 41.5| 704.2|55.2|58.5|0.99|0.98|0.99|0.98|
> |ERU|`26.2`|`30.5`|`40.5`|`708.8`|`43.2`|`50.6`|`0.59`|`0.56`|`0.74`|`0.53`|
> |w/o RFAT|**27.1**|**32.1**|**41.5**|**710.5**|**44.3**|**52.3**|**0.62**|**0.57**|**0.76**|**0.54**|
> |w/o ERM|25.7 |29.8| 40.2| 709.8 |42.5 |51.5|0.57|0.56|0.72|0.53|
>
> `Ablation results of Unlearning Robustness`
>
> |Method|RWKU|RWKU|RWKU|WMDP|WMDP|
> |:--|:--:|:--:|:--:|:--:|:--:|
> ||FB↓|QA↓|AA↓|AccBio↓|AccCyber↓|
> |Original|51.9|46.8|57.5|63.2|42.8|
> |ERU|**29.2**|**27.1**|**25.5**|**24.8**|**28.4**|
> |w/o RFAT|44.4|39.6|46.8|51.4|37.9|
> |w/o ERM|`29.8`|`28.6`|`26.1`|`26.5`|`30.4`|
>
> By analyzing the above three tables, it can be seen that after removing the two core components of ERU, it shows significant differences in the three dimensions of unlearning (Unlearning Effectiveness, Utility Preservation, Unlearning Robustness). The best method is **bolded**, and the second-best method in `highlight`.
>
> Specifically, removing RFAT does not significantly affect Unlearning effectiveness, but improves utility Preservation(Rea: 26.2 $\to$ 27.1, KnowMem: 43.2 $\to$ 44.3), but causes a near collapse of robustness (FB:29.2 $\to$ 44.4, AccBio:24.8 $\to$ 51.4).  This indicates that there is an inherent contradiction between adversarial training and model utility, and seeking a balance between them is the key to construct a robust unlearning mechanism.
>
> The removal of the elastic reward setting directly weakened the Unlearning Effectiveness (FB: 29.2$\to $33.4, QA: 27.1$\to $31.2), confirming the core supporting role of this component for Unlearning Effectiveness.  At the same time, due to the failure of early-stage gradient weight smoothing (see Appendix D.2 for details), its removal also impairing the performance of Utility Preservation.
>
> *****
>
> > **Weakness and Question (2.1)**: Figure 6 (PS: It does not have a caption) shows the effect of alpha and shows that setting alpha away from zero will improve the performance.
>
> Thank you for the reminder. To clarify, Figure 6 illustrates the impact of different $\alpha$ (the hyperparameter controlling the influence of the reference model in our elastic reward setting) on unlearning effectiveness across two LLMs: LLaMA-2-7B-Chat and LLaMA-3-8B-Instruct.
>
> *****
>
> > **Weakness and Question (2.2)**: It seems that the optimal alpha highly depends on the model (what about across datasets? What are the optimal alpha?) Is there anyway to explain this?
>
> Your observation that the optimal α depends on the model is correct and consistent with our findings. We explain this below.
>
> According to the loss function of NPO (Equation 5 in the paper), as unlearing proceeds in the expected direction, the prediction probability of the current model $\pi_\theta$ for the knowledge to be forgotten will continue to decrease and gradually deviate from the prediction probability of the reference model $\pi_{ref}$ for this part of the knowledge. This is the basic principle of reference-based reward for unlearing. Correspondingly, the reference-free reward(Equation 12 in the paper) directly replaces the "ruler" role of the reference model in a rigid way by uniformly distributing $U (y | x)$.
>
> The parameter $\alpha$  in the elastic reward setting we proposed is precisely the valve that balances and regulates these two "Ruler". When $\alpha$ decreases, the elastic reward setting tends towards reference-free reward. Conversely, it tends towards reference-based reward. Therefore, if users have sufficient trust in the reference model and believe that it can accurately provide a large enough prediction probability for forgetting knowledge during the unlearing process, it is beneficial to appropriately increase $\alpha$ to bias the reference-based reward. This is why in **Figure 6(b)** of the paper, with the increase of $\alpha$, the Unlearing Effectiveness is better (with a lower accuracy rate in the forget set). On the contrary, when the reference model is incompetent (LLaMA 2-7B-Chat vs. LLaMA 3-8B-Instruct), it is necessary to reduce reliance on the reference model and lean more towards reference-free rewards (Figure 6(a) in the paper).
>
> In addition, your concern about the influence of different datasets on the optimal α is also worth delving into. We have supplemented the experiments on multiple benchmarks as follows:
>
> `LLaMA-2-7B-Chat`
> |Method|RWKU|RWKU|RWKU|TOFU|TOFU|
> |:--|:--:|:--:|:--:|:--:|:--:|
> ||FB↓| QA↓ |AA↓|Forget05-FQ↑ |Forget10-FQ↑|
> |α=0.00|33.8| 31.1|35.6| 0.92 |0.47 |
> |α=0.02|32.0| 30.8|31.8| 0.85 |0.45 |
> |α=0.05|29.2| 27.1|25.5| 0.73 |0.48 |
> |α=0.10|30.9| 27.4|27.3| 0.69 |0.46 |
> |α=0.15|31.3| 28.1|28.5| 0.68 |0.42 |
> |α=0.20|31.5| 28.2|26.7| 0.65 |0.45 |
>
> `LLaMA-3-8B-Instruct`
> |Method|RWKU|RWKU|RWKU|TOFU|TOFU|
> |:----|:--:|:--:|:--:|:--:|:--:|
> ||FB↓| QA↓ |AA↓|Forget05-FQ↑ |Forget10-FQ↑|
> |α=0.00|33.1| 23.4 |23.1| 0.62| 0.49 |
> |α=0.02|33.4| 22.6 |21.6| 0.59| 0.48 |
> |α=0.05|32.2| 22.8 |20.9| 0.68| 0.50 |
> |α=0.10|32.5| 22.4 |19.2| 0.73| 0.49 |
> |α=0.15|31.8| 21.6 |19.4| 0.78| 0.52 |
> |α=0.20|31.2| 21.1 |18.5| 0.84| 0.56 |
>
> From the table, we can see that the trends of most datasets are basically consistent with the aforementioned analysis. However, there are differences in specific instances (such as Lama-2-7B-chat in RWKU), which may be due to the high prediction probability of the model for forget knowledge in this dataset. Therefore, we conclude that although both the model and the dataset can influence the optimal value of α, the essence still depends on the prediction accuracy of the model for forget knowledge.
>
> *****
>
> > **Weakness and Question (2.3)**: Do you have any more intuition about what the hyperparameter alpha is and what it controls? And the most importantly, in what case would you expect a larger optimal alpha?
>
> As we replied in **Weakness and Question (2.2)**, when the reference model can output a relatively high prediction probability for forget knowledge, the $\alpha$ can be correspondingly increased. On the contrary, if the reference model $\pi_{ref}$ cannot provide a large enough prediction probability for the forget knowledge, and the prediction probability ratio of the current model $\pi_\theta$ and the reference model $\pi_{ref}$ is too large, the ERU loss function (see Equation 16 of the paper) will be difficult to converge, and $\alpha$ should be reduced to enhance the effect of uniform distribution $U (y | x)$. In addition, too high $\alpha$ will make the model utility decline rapidly in the early stage of unlearning as in NPO, resulting in over-unlearning. Based on the above considerations, we suggest setting $\alpha$ within the smaller range of [0, 0.2], which has been verified in the experiments of the paper.
>
> ### **Concluding remarks.**
>
> We would be grateful if you could let us know whether our explanations have addressed your concerns. Please let us know if you have any other questions or concerns.

---

> > ### Author Response · Authors · 2025-08-06
> >
> > Dear reviewer NnGx,
> >
> > Thanks for taking the time to review our work, We have carefuly considered your comments and made every effort to respond to your concerns.
> >
> > If you have any further questions or require additional clarification, please kindly let us know.
> >
> > Best regards.

---

### Official Review · Reviewer_gzkG · 2025-07-09

**Clarity:** 2
**Significance:** 2
**Originality:** 3
**Rating:** 3
**Confidence:** 4

**Summary:**

This paper proposes Elastic Robust Unlearning. This method combines an "elastic reward" which regularizes the unlearning algorithm by adding a parameter to vary the influence of the reference model, with the technique of refusal feature adversarial training to make the model more robust to relearning and jailbreaking attacks.

**Questions:**

- Why is TOFU not included in the utility analysis in 4.4? It does not seem to appear in the appendix either.
- The ablation study in table 8 is a useful study but the results are presented in a confusing way. Why is unlearning effectiveness evaluated on a different dataset compared to unlearning robustness? Ideally the ablation would be on a single consistent benchmark.
- How do the regularized ERU models (in table 2) perform on the forget tasks (table 1)? This is key to understanding the forget/retain tradeoff - otherwise the comparisons are apples and oranges.

**Ethical Concerns:**

["NO or VERY MINOR ethics concerns only"]

**Final Justification:**

I appreciate the authors' response. Due to the concerns listed in my responses, I have maintained my score.

**Limitations:**

Yes

**Quality:**

2

**Strengths And Weaknesses:**

Strengths:
- The method comprehensively covers unlearning effectiveness as well as robustness, incorporating recent techniques to improve performance on both.
- The evaluation thoroughly covers several benchmarks that are currently state of the art.
- The proposed method of an "elastic" reward is interesting and intuitive, to interpolate between the reference-based and reference-free setting.

Weaknesses:
- The results seem generally inconclusive. ERU does not do uniformly better than other methods across benchmarks, and as such it seems to be another point on the forget/retain tradeoff curve rather than pushing the Pareto frontier forward.
- The elastic reward is interesting but the reasoning for it seems fairly heuristic. NPO itself is a version of regularized gradient ascent, and the elastic reward seems to be an alternative regularizer that incorporates the reference model. The arguments for why this should lead to better unlearning seem to be based on a fairly hand-wavy explanation of training dynamics (Appendix B). As such it's hard to predict when this method will be effective.
- Some results seem cherry picked: why is ERU regularized not included in Table 1? Why is WMDP evaluated only on bio and cyber, but not chemistry? Why is TOFU not included in Table 2? Additionally, presenting forget and retain in separate tables rather than side by side makes it difficult to determine which methods perform the best.

---

> ### Author Rebuttal · Authors · 2025-07-28
>
> Dear reviewer gzkG
>
> Thank you for a thoughtful and constructive review. We are pleased to hear your positive assessment of the novelty of our work and think our "elastic" reward is interesting and intuitive. We hope to address your concerns and questions in our response below.
>
> ###  **Weaknesses:**
>
> > **Weakness (1)** : The results seem generally inconclusive. ERU does not do uniformly better than other methods across benchmarks [...]
>
> Most baseline methods (except advNPO) only deal with the two-dimensional trade-off of unlearning effectiveness and model utility (such as GA sacrificing utility for unlearning effectiveness). By jointly optimizing three key dimensions, namely unlearning effectiveness, utility preservation and unlearning  robustness, ERU achieves a coordinated improvement among them and advances the Pareto boundary. ERU has significant advantages in any dimension:
>
> `Unlearning Effectiveness`: The results in table 1 of the paper show that ERU significantly outperforms suboptimal methods on three of the four datasets (RWKU,TOFU,WMDP). Only in the MUSE-News dataset is it inferior to GA. However, it should be pointed out that GA achieves this advantage at the cost of significantly reducing the model's utility.
>
> `Utility Preservation`:  It can be seen from the results in table 2 of the paper that the original ERU outperforms the baseline unlearning method without regularization in most evaluation metrics, and some metrics even approach or exceed the methods enhanced by regularization. Taking the KnowMem metric as an example, ERU's score (43.2) surpassed $GA_{KLR}$ (41.8), $NPO_{GDR}$ (40.5), and was close to $NPO_{KLR}$(46.4). Particularly, when ERU also incorporates the regularizer, its utility maintenance ability is significantly improved, specifically manifested as the outstanding improvement of $ERU_{KLR}$ on the MMLU dataset and $ERU_{GDR}$ on the MUSE-News dataset.
>
> `Unlearning Robustness`: As can be seen from the results in Figure 2 and Figure 3 of the paper, ERU significantly outperforms the suboptimal method in terms of Unlearning Robustness. For example, under 1,000 samples of fine-tuned retraining attacks, ERU can still maintain 83% unlearning performance, while the suboptimal method advNPO can only maintain 72%.
>
> Furthermore, in terms of time efficiency, ERU saves approximately half of the training time compared to the advNPO method, which also falls within the category of adversarial training.
>
> > **Weakness (2)** : The elastic reward is interesting but the reasoning for it seems fairly heuristic. NPO itself is a version of regularized gradient ascent [...]
>
> The elastic reward we proposed is not a heuristic design but a principle-based solution to address the limitations of the Rigid Reward Setting (Appendix B): reference-based reward can lead to instability in early training, while reference-free reward lose instance-specific signals.   Therefore, we redefined the reference model as the joint reference model, unifying the reference-based and reference-free rewards (Equation 13).   Then, after substituting it into the objective function of NPO, we can derive the objective function of EU (Appendix D.1).    We derived the adaptive adaptive smoothing weight from the objective function of EU through gradient analysis (Equation 37) and theoretically proved that it can avoid early-stage gradient weight smoothing ineffective (Appendix D.2).
>
> > **Weakness (3.1)** : Some results seem cherry picked: why is ERU regularized not included in Table 1? Why is WMDP evaluated only on bio and cyber, but not chemistry?
>
> Thank you for your valuable suggestions! We promise All experiments follow established protocols from RWKU, MUSE, TOFU, and WMDP benchmark. Since our experimental setup on WMDP followed simNPO [1], we only covered bio and cyber. In order to address concerns, we now supplement the regularized ERU experimental results and the experimental results of the chemistry on WDMP benchmark as follows:
>
> |Method|RWKU|RWKU|RWKU|MUSE-News|MUSE-News|MUSE-News|TOFU|TOFU|WMDP|WMDP|WMDP|
> |  :----  | :----: | :----: | :----: | :----: | :----: | :----: | :----: | :----: | :----: | :----: | :----: |
> ||FB↓| QA↓ |AA↓|VerbMem↓ |KnowMem↓ |PrivLeak |Forget05-FQ↑| Forget10-FQ↑|AccBio↓ | AccCyber↓|  AccChem↓|
> |Original|51.9 | 46.8|  57.5| 58.3| 63.7| -99.8| 3.2e-16| 2.1e-19| 63.2| 42.8|52.4|
> |DPO|38.9 | 40.7 | 41.5| 33.2| 37.2| 109.6| 1.2e-4| 3.5e-7| 28.9| 33.5|35.2|
> |IDK|40.5|  40.6 | 45.4| 35.6| 39.1| 104.3| 4e-5| 5e-8| 29.3| 34.2| 34.8|
> |GA|44.5 | 39.6|  47.3| **0.0**| **0.0**| 20.8| 0.05| 8.1e-10| 37.4| 30.1|31.4|
> |GradDiff|46.4|  42.2|  48.6| 25.9| 31.0| 105.3| 0.09| 7.9e-3| 38.6| 33.5|32.7|
> |GA_KLR|46.8 | 41.4|  44.3| 27.4| 58.6| -51.6| 0.11| 3.4e-5| 37.9| 33.2| 33.5|
> |NPO|33.6 | 31.3| 32.8| 10.8| 13.4| 30.4| 0.66| 0.19| 29.6| 32.7| 31.8|
> |NPO_GDR|34.8|  34.7|  38.1| 13.2| 48.6|101.3| 0.44| 0.24| 31.8| 33.0| 32.2|
> |NPO_KLR|37.6|  34.5 | 38.5| 16.6| 38.6| -56.7| 0.43| 0.17| 32.4| 32.9| 32.4|
> |SimNPO|34.2 | 31.8 | 37.5| 12.6| 11.3| 14.9| **0.97**| `0.45`| 28.6| 29.8|31.7|
> |AdvNPO|35.9 | 33.2|  **25.2** | 13.7 | 12.8 | 24.6 | 0.63 | 0.26|  29.9 | 33.2| 32.8|
> |ERU|**29.2**| **27.1**| `25.5`| `10.4` |`9.2` |**12.3**| `0.73`| **0.48** |**24.8**| **28.4**|**27.2**|
> |ERU_GDR|29.8| 28.6| 25.9| 11.4 |10.8 |`14.1`| 0.69| `0.45` |`25.7`| `29.4`|28.6|
> |ERU_KLR|`31.4`| `27.9`| 27.5| 11.6 |11.2 |15.3| 0.67| 0.42 |27.5| 30.2|`28.2`|
>
> From the table, we can see that although the regularizer will slightly hurt the ERU unlearning effectiveness of ERU (this trend is consistent with other methods), it can still maintain good performance. The best method is **bolded**, and the second-best method  in `highlight`.
>
> > **Weakness (3.2)** : Why is TOFU not included in Table 2?
>
> Thank you for pointing out this oversight. We now supplement the experimental results of the TOFU dataset in the utility preservation dimension as follows:
>
> |Method|RWKU|RWKU|RWKU|RWKU|MUSE-News|MMLU|TOFU-Forget05|TOFU-Forget05|TOFU-Forget10|TOFU-Forget10|
> |  :----  | :----: | :----: | :----: | :----: | :----: | :----: | :----: | :----: | :----: | :----: |
> ||Rea↑ |Tru↑ |Fac↑ |Flu↑| KnowMem↑|Accuracy↑|Probability↑|ROUGE↑|Probability↑|ROUGE↑|
> |Original|26.9 |30.4| 41.5| 704.2|55.2|58.5|0.99|0.98|0.99|0.98|
> |DPO|26.4 |25.2 |32.4|`710.6`|32.8|46.8|`0.74`|0.53|0.76|0.54|
> |IDK|26.8|27.9| 36.7|**712.5**|37.3|50.2|**0.76**|0.55|`0.78`|0.55|
> |GA|25.8| 30.7 |40.2 |707.6|0.0|48.3|0.00|0.00|0.00|0.00|
> |GradDiff|24.8  |30.4  |`41.1`|707.5 |27.3 |51.2|0.49|0.42|0.57|0.48|
> |GA_KLR|26.2 |29.8 |40.6 |708.3|41.8|51.8|0.48|0.44|0.53|0.49|
> |NPO|26.2 |30.5|`41.1`| 694.6 |27.5 |47.6|0.51|0.47|0.46|0.44|
> |NPO_GDR|26.5 |30.4| 40.8| 705.2|40.5|51.7|0.56|0.55|0.65|0.53|
> |NPO_KLR|26.3|**31.2**|40.9| 703.8|`46.4`|50.5|0.56|0.54|0.71|0.55|
> |SimNPO|26.3 |29.4| 40.5| 691.3|43.5|50.2|0.56|0.54|0.72|0.53|
> |AdvNPO|24.3 |26.5| 39.8 |672.8|24.3|41.2 |0.48|0.46|0.46|0.45|
> |ERU|26.2 |30.5| 40.5| 708.8 |43.2 |50.6|0.59|`0.56`|0.74|0.53|
> |ERU_GDR|26.1 |30.7| **41.2**| 707.5 |**47.2**|`52.1`|0.72|**0.57**|**0.79**|**0.56**|
> |ERU_KLR|**26.6** |`30.8`| 40.9| 708.6 |44.2 |**53.4**|`0.74`|**0.57**|`0.78`|`0.55`|
>
>
> From the table, we can see that the regularizer can consistently enhance the model utility of ERU and outperforms the suboptimal method. The conclusion is consistent with the description in the paper.
>
> > **Weakness (3.3)** :Additionally, presenting forget and retain in separate tables rather than side by side makes it difficult to determine which methods perform the best.
>
> Since our original intention was to discuss the performance of the unlearing method from three dimensions respectively, to avoid confusion, the corresponding experimental tables and images are also presented separately. To present the advantages of ERU more clearly, we will present the comprehensive results in the appendix.
>
> ###  **Questions:**
>
> >  **Question (1):** :Why is TOFU not included in the utility analysis in 4.4? It does not seem to appear in the appendix either.
>
> Please refer to our response to **Weakness (3.2)**.
>
> >  **Question (2):** The ablation study in table 8 is a useful study but the results are presented in a confusing way. [...]
>
> Given that in the discussion of unlearing robustness in Section 4.3, we demonstrated the unlearing robustness of each method by evaluating  the change of unlearing effectivenes on WMDP, we have followed this metric here. We apologize for any confusion this may have caused. To address your concerns, we now supplement the experimental results of unlearing robustness on RWKU as follows:
>
> |Method|Unlearning Effectiveness|Unlearning Effectiveness|Unlearning Effectiveness|Unlearning Robustness|Unlearning Robustness|Unlearning Robustness|Utility Preservation|
> |  :----  | :----: | :----: | :----: | :----: | :----: | :----: | :----: |
> ||RWKU|RWKU|RWKU|RWKU|RWKU|RWKU|MMLU|
> ||FB↓|QA↓|AA↓|FB↓|QA↓|AA↓|Accuracy↑|
> |Original|51.9|46.8|57.5|51.9|46.8|57.5|58.5|
> |ERU|29.2|27.1|25.5|29.2|27.1|25.5|50.6|
> |w/o RFAT|29.6|28.4|35.8|44.4 (15.2↑)|39.6 (12.5↑)|46.8 (21.3↑)|52.3|
> |w/o ERM|33.4| 31.2|32.5|29.8| 28.6|26.1|51.5|
>
> From the table, we can see that after removing the refusal feature adversarial training (RFAT), the unlearing effectiveness of the unlearned model is significantly impaired (FB:29.2$\rightarrow$44.4, QA:27.1$\rightarrow$39.6, AA:25.5$\rightarrow$46.8), which highlights the importance of RFAT.
>
> >  **Question (3):** How do the regularized ERU models (in table 2) perform on the forget tasks (table 1)? This is key to understanding the forget/retain tradeoff [...]
>
> Please refer to our response to **Weakness (3.3)**.
>
> ### **Concluding remarks.**
>
> We would be grateful if you could let us know whether our explanations have addressed your concerns. Please let us know if you have any other questions or concerns.
>
> ### **References.**
>
> [1] Fan , et al. (2024) Simplicity prevails: Rethinking negative preference optimization for llm unlearning.

---

> ### Comment · Reviewer_gzkG · 2025-08-05
>
> I appreciate the authors' work to report more extensive evaluation results. As of now, I will prefer to maintain my score.
>
> In particular, while the authors do analyze training dynamics and the impact on model utility, it is still unclear to me *why* this algorithm achieves the point in the tradeoff space that it does. There are hints toward an understanding in Appendix B and lines 630-633 of the paper but these are still far from a clear explanation of how the improved training dynamics lead to more effective unlearning.
>
> I do however appreciate the empirical effectiveness of the method and note that other reviewers have raised their scores. I am happy to discuss further during the reviewer-AC period and defer to the greater consensus.

---

> ### Author Response · Authors · 2025-08-06
>
> Dear reviewer gzkG,
>
> We sincerely thank the reviewer for their constructive feedback and acknowledgment of ERU’s empirical strengths. We are very willing to clarify how the design of ERU achieves its performance trade-offs.
>
> **Why can ERU achieve a good unlearning-utility trade-off**
>
> The unlearning method with rigid reward setting has demonstrated its unlearning effectiveness[1][2]. **We inherit its excellent unlearning effectiveness through the elastic reward setting (Equation 23) and seek the smallest possible utility loss.** The elastic reward setting of ERU specifically **alleviates the over-unlearning phenomenon** that occurs in the early stage of training (Appendix B), avoids excessive loss of model utility in the early stage of unlearning, and does not completely lose the role of the reference model. Through the theoretical derivation of gradient analysis (Appendix D.2) and the experimental verification of ERU (Table 1 and Table 2), we have confirmed its ability to achieve a good trade-off.
>
>
> **A clearer understanding of Appendix B and D.2**
>
> In Appendix B, we analyze the reasons for the low model utility of the previous method through gradient analysis (Figure 4), and strive to avoid the situation of Equation 22.
>
> $W^{init}\_{\theta}(x, y)=\frac{2 \pi_{\theta}^{\beta}(y \mid x)}{\pi_{\theta}^{\beta}(y \mid x)+\pi_{\mathrm{ref}}^{\beta}(y \mid x)}\approx 1$ (Equation 22)
>
> Equation 37 in Appendix D.2 indicates that the elastic reward setting ensures that its weight gradient is no longer approximately 1 during the early training period but is related to the response length (Equation 37). The ablation experiment in Appendix F.3 further indicates that removing the elastic reward will lead to the failure of the overall trade-off.
>
> We hope this structured explanation can clarify how the design of ERU achieves good performance in the trade-off space. We're happy to further discuss any remaining questions during the discussion period.
>
>
>
>
> [1] Fan , et al. (2024) Simplicity prevails: Rethinking negative preference optimization for llm unlearning.
>
> [2] Zhang , et al. (2024) Negative Preference Optimization: From Catastrophic Collapse to Effective Unlearning.

---

> > ### Author Response · Authors · 2025-08-09
> >
> > Dear reviewer gzkG,
> >
> > Thank you for your valuable discussion about our work, we have made every effort to respond to your concerns.
> >
> > As the discussion period is coming to an end, if you have any further questions or need additional clarification, please let us know.
> >
> > Best regards,
> >
> > The authors of Paper 8095

---

### Note · Authors · 2025-08-13

Dear NeurIPS 2025 Reviewers, AC, SAC, and PC,

We thank all reviewers for their thoughtful reviews, valuable suggestions, and for taking the time to read our paper! Meanwhile, we also appreciate the responsibility and tireless efforts of AC, SAC and PC.

We particularly appreciate the positive recognition of many aspects of our work, including its novelty (gzkG, NnGx, uCL3), significance (uCL3, pWXE), empirical comparison and experimental setup (gzkG, NnGx).
We hope we have addressed all questions and concerns raised by the reviewers and are happy to discuss any remaining concerns or questions during the rebuttal.

**Main Supplement:**

1. Improving the clarity of the text with additional explanations, including explanations of the ablation experiment and a deeper understanding of the elastic reward setting.      (Rebuttal to Reviewer gzkG, NnGx, uCL3, pWXE)

2. Through more experiments to verify the effectiveness of ERU, Including the unlearing effectiveness of regularized ERU, the experimental results of the chemistry on WDMP benchmark, and the utility analysis on TOFU.  (Rebuttal to Reviewer gzkG)

3. Provide more analysis of parameter alpha.  (Rebuttal to Reviewer NnGx)

4. Provide more details on balancing the inner and outer optimization.  (Rebuttal to Reviewer uCL3)

5. Added a discussion on the robustness of adaptive attacks.  (Rebuttal to Reviewer uCL3)

6. Under the same elastic reward setting, combines ERU with other sophisticated adversarial training techniques for comparison to verify the advantages of refusal feature adversarial training (RFAT).  (Rebuttal to Reviewer pWXE)

7. Perform a statistical significance test to demonstrate that the proposed method yields significant improvements in unlearning effectiveness while preserving model utility.  (Rebuttal to Reviewer pWXE)

We would be grateful if you could let us know whether our explanations have satisfactorily addressed your concerns.  We are also open to discussing any other questions you may have.

Best regards,

The Authors of Paper 8095

---

### Decision · Program_Chairs · 2025-09-17

**Decision:**

Accept (poster)

**Comment:**

This paper introduces elastic robust unlearning (ERU), with two main contributions: (1) an elastic reward setting that interpolates between reference-based and reference-free rewards, and (2) refusal feature ablation and adversarial training. The problem of robust unlearning in LLMs is timely and important, and the design of ERU is well-motivated. Empirical evaluations are extensive, covering multiple benchmarks, and additional ablations and significance tests provided in the rebuttal further strengthened the case. Several reviewers highlighted the strong empirical results.

On the weakness side, the main contention lies in the novelty and theoretical justification of the elastic reward setting. During the discussion, one reviewer remained unconvinced, viewing it as little more than a heuristic interpolation between existing rigid schemes and finding the analysis insufficiently compelling, leading to a borderline stance. In contrast, other reviewers, while less active in discussion, evaluated the work more positively. The AC considers the combination of the elastic reward with refusal feature ablation to constitute a meaningful contribution overall, although the novelty concerns are valid: E.g., AC found the following work relevant to robust unlearning: Fan, et al. "Towards llm unlearning resilient to relearning attacks: A sharpness-aware minimization perspective and beyond." arXiv preprint arXiv:2502.05374 (2025).

Balancing these perspectives, I find the empirical strength of ERU outweighs the novelty concern. This work makes a useful contribution to the growing area of machine unlearning in LLMs.